# Using atmospheric observations to quantify annual biogenic carbon dioxide fluxes on the Alaska North Slope

Luke D. Schiferl[1,2], Jennifer D. Watts[3], Erik J. L. Larson[4], Kyle A. Arndt[3,5,6], Sébastien C. Biraud[7], Eugénie S. Euskirchen[8], Jordan P. Goodrich[5,9], John M. Henderson[10], Aram Kalhori[5,11], Kathryn McKain[12,13], Marikate E. Mountain[10], J. William Munger[2], Walter C. Oechel[5,14], Colm Sweeney[12], Yonghong Yi[15,16], Donatella Zona[5,17], and Róisín Commane[1,18]

[1]Lamont-Doherty Earth Observatory, Columbia University, Palisades, New York, USA.
[2]Harvard John A. Paulson School of Engineering and Applied Sciences, Cambridge, Massachusetts, USA.
[3]Woodwell Climate Research Center, Falmouth, Massachusetts, USA.
[4]Department of Organismic and Evolutionary Biology, Harvard University, Cambridge, Massachusetts, USA.
[5]Department of Biology, San Diego State University, San Diego, California, USA.
[6]Earth Systems Research Center, Institute for the Study of Earth, Oceans, and Space, University of New Hampshire, Durham, New Hampshire, USA.
[7]Lawrence Berkeley National Laboratory, Berkeley, California, USA.
[8]Institute of Arctic Biology, University of Alaska Fairbanks, Fairbanks, Alaska, USA.
[9]Ministry for the Environment, Wellington, New Zealand.
[10]Atmospheric and Environmental Research, Inc., Lexington, Massachusetts, USA.
[11]GFZ German Research Centre for Geosciences, Potsdam, Germany.
[12]Global Monitoring Laboratory, Earth System Research Laboratories, NOAA, Boulder, Colorado, USA.
[13]Cooperative Institute for Research in Environmental Sciences, University of Colorado, Boulder, Colorado, USA.
[14]Department of Geography, University of Exeter, Exeter, United Kingdom.
[15]Joint Institute for Regional Earth System Science and Engineering, University of California, Los Angeles, California, USA.
[16]College of Surveying and Geo-Informatics, Tongji University, Shanghai, China.
[17]Department of Animal and Plant Sciences, University of Sheffield, Western Bank, Sheffield, United Kingdom.
[18]Department of Earth and Environmental Sciences, Columbia University, New York, New York, USA.

*Correspondence to*: Luke D. Schiferl (schiferl@ldeo.columbia.edu)

**Abstract.** The continued warming of the Arctic could release vast stores of carbon into the atmosphere from high-latitude ecosystems, especially from thawing permafrost. Increasing uptake of carbon dioxide ($CO_2$) by vegetation during longer growing seasons may partially offset such release of carbon. However, evidence of significant net annual release of carbon from site-level observations and model simulations across tundra ecosystems has been inconclusive. To address this knowledge gap, we combined top-down observations of atmospheric $CO_2$ concentration enhancements from aircraft and a tall tower, which integrate ecosystem exchange over large regions, with bottom-up observed $CO_2$ fluxes from tundra environments and found that the Alaska North Slope is not a consistent net source or net sink of $CO_2$ to the atmosphere (ranging from –6 to +6 TgC yr$^{-1}$ for 2012–2017). Our analysis suggests that significant biogenic $CO_2$ fluxes from unfrozen terrestrial soils, and likely inland waters, during the early cold season (September–December) are major factors in determining the net annual carbon balance of the North Slope, implying strong sensitivity to the rapidly warming freeze-up period. At the regional level, we find no evidence for previously reported large late cold season (January–April) $CO_2$ emissions to the atmosphere during the study

period. Despite the importance of the cold season $CO_2$ emissions to the annual total, the interannual variability of the net $CO_2$ flux is driven by the variability in growing season fluxes. During the growing season, the regional net $CO_2$ flux is also highly sensitive to the distribution of tundra vegetation types throughout the North Slope. This study shows that quantification and characterization of year-round $CO_2$ fluxes from the heterogeneous terrestrial and aquatic ecosystems in the Arctic using both site-level and atmospheric observations is important to accurately project the earth system response to future warming.

## 1 Introduction

The Arctic surface air temperature is warming at twice the rate of the global average (Box et al., 2019; Meredith et al., 2019). Continued thawing of Arctic permafrost has the potential to release vast stores of carbon into the atmosphere, thereby further accelerating warming (Schuur et al., 2015; Hugelius et al., 2014). In the biosphere, the net $CO_2$ flux is the balance between uptake of $CO_2$ by vegetation through photosynthesis (negative net $CO_2$ flux indicates removal from the atmosphere) and release of $CO_2$ into the atmosphere by plant and microbial respiration (positive net $CO_2$ flux indicates a source to the atmosphere). Arctic growing seasons are short (~3 months), and the long, cold season dominates the seasonal cycle. The transition between the growing and cold seasons is marked by the soil zero-curtain period, when belowground temperatures of the active layer above frozen permafrost remain near freezing; the active layer is insulated by snow and ice at the surface and warmed by the latent heat release of freezing water (Outcalt et al., 1990). During the zero-curtain period, soil respiration can remain active in deeper soils for weeks to months after the end of the growing season (Zona et al., 2016; Romanovsky and Osterkamp, 2000). As the climate warms, the active layer above permafrost deepens, thawed soils become wetter, a larger volume of soil remains unfrozen for a longer period of time, and the duration of the zero-curtain period plays an increasingly important role in determining the net carbon exchange in Arctic ecosystems (Kim et al., 2012; Arndt et al., 2019). Recent work has shown a significant cold season source of $CO_2$ from Arctic ecosystems, including more than 70% increase in October–December $CO_2$ concentration enhancements in the past 40 years, consistent with an increase in cold season respiration, which is not well represented in earth system models (Commane et al., 2017; Natali and Watts et al., 2019). Neglecting these processes could lead to large underestimation of $CO_2$ emissions, biasing current and future climate projections.

Tundra ecosystems, characterized by frozen soils covered in low shrubs, sedges, grasses, and mosses, make up approximately 50% of the Arctic landscape (Raynolds et al., 2019). Lacking trees, the magnitude of net $CO_2$ uptake in tundra during the growing season is relatively small and may be offset by emissions from respiration that can continue well into the cold season (Watts et al., 2021). In the past, year-round $CO_2$ flux measurements from tundra ecosystems were rare due to difficulties in maintaining instrumentation under remote and extreme cold conditions (Euskirchen et al., 2017; Kittler et al., 2017; Goodrich et al., 2016). Long-term year-round $CO_2$ concentration measurements have been made in the Arctic at a small number of tall towers, which have been situated to sample clean marine air off the ocean (Jeong et al., 2018; Worthy et al., 2009). While aircraft provide greater spatial coverage over land than these towers, they tend to operate for short durations, and their temporal coverage is limited by weather and visibility during the cold season (Chang et al., 2014; Commane et al., 2017;

Miller et al., 2016). However, the recent increase in availability of observations of gas fluxes and concentrations within a
particular tundra region, the Alaska North Slope (Fig. 1a), is making it possible to better conduct year-round multi-scale
assessments of tundra ecosystems, with the aim of improving our understanding of $CO_2$ sink/source activity and carbon budgets
in these environments.

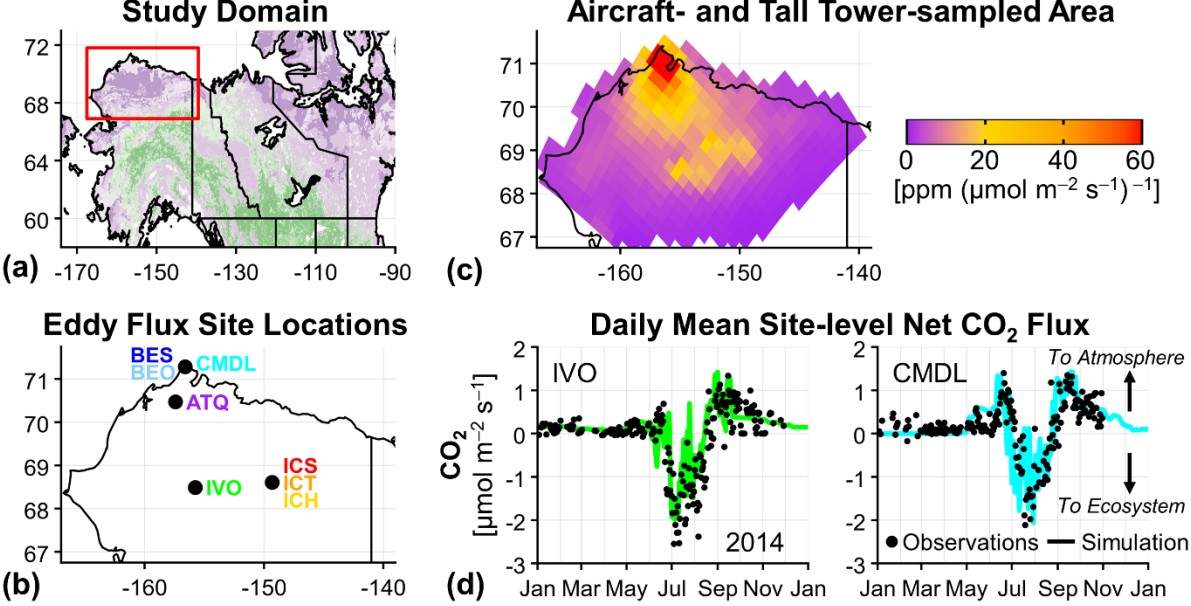


**Figure 1.** Alaska North Slope study region, eddy flux site locations, area sampled by aircraft and tower, and example results from the eddy
flux site measurement-model comparison. **(a)** North Slope region (red box) within Alaska and northwestern Canada. Tundra areas shown in
purple and boreal forest areas shown in green (Luus et al., 2017). **(b)** Location of eddy flux measurement sites on the Alaska North Slope
used in this analysis. **(c)** Ten-day WRF-STILT footprints used to sample $CO_2$ flux models, summed for all aircraft and tall tower $CO_2$
observations used in this analysis. Colors represent values greater than 0 and are saturated at 60 ppm (µmol m$^{-2}$ s$^{-1}$)$^{-1}$. Maximum value near
Utqiaġvik, Alaska is 324 ppm (µmol m$^{-2}$ s$^{-1}$)$^{-1}$. **(d)** Timeseries of observed (black dots) and simulated (colored lines) site-level daily mean
net $CO_2$ flux for 2014 at IVO (left) and CMDL (right) eddy flux measurement sites, where site-level TVPRM net $CO_2$ flux simulations are
driven by NARR meteorology and the CSIF SIF product. Positive net $CO_2$ flux values indicate $CO_2$ fluxes into the atmosphere throughout
this study. A comparison for all eight eddy flux sites is provided in Fig. S1 in Supplement.
Currently, observations and models do not agree on the sign of the annual net $CO_2$ flux across the Alaska North Slope
region. Site-level measurements and atmospheric observations suggest this region is a net $CO_2$ source (Commane et al., 2017;
Oechel et al., 2014; Euskirchen et al., 2017). However, a comparison of process-based models of the North Slope found large
variability in the sign and magnitude of the net $CO_2$ flux with an approximately neutral regional annual net $CO_2$ flux multi-
model mean of –3.5 ± 67 TgC yr$^{-1}$ (Fisher et al., 2014). In a more recent study, Tao et al. (2021) found an annual net $CO_2$ flux
range of –9 to 12 TgC yr$^{-1}$ for the years 2010–2016, with only 2014 being an annual net $CO_2$ source. Extrapolating from site-
level $CO_2$ flux measurements to regional budgets is difficult due to the extreme heterogeneity of tundra ecosystems in the
North Slope and a lack of spatial and seasonal representativeness by existing flux monitoring sites (Pallandt et al., 2022).
In this study, we compare *bottom-up* flux estimates with *top-down* atmospheric observations from aircraft and a tall
tower using an integrated modeling approach to quantify the $CO_2$ budget sign and magnitude of the Alaska North Slope. Our
framework first applies a bottom-up approach to understand Arctic tundra ecosystem $CO_2$ fluxes, constrained by site-level
observations, using an empirical model ensemble of $CO_2$ fluxes derived from eddy flux measurements representing varied
tundra ecosystems within the region. We then apply top-down information gained from regional $CO_2$ concentration
enhancement observations measured by a tall tower and aircraft, which sample the atmosphere-biosphere exchange throughout
the Alaska North Slope, to evaluate the range of potential $CO_2$ fluxes identified by the bottom-up model ensemble for 2012–
2017. This evaluation also identifies the ecosystem parameterizations, vegetation distributions, and environmental drivers that
best characterize the observed spatial and temporal distribution of biogenic $CO_2$ in the atmosphere across the region. By
developing regional $CO_2$ budgets constrained by both atmospheric observations and ecosystem environmental responses, we
can better project how Arctic tundra ecosystems will respond to climate change on annual and decadal timescales.
**2 Materials and methods**
**2.1 Observed $CO_2$ concentrations and fluxes on the Alaska North Slope**
**2.1.1 Atmospheric $CO_2$ concentration observations**
We use a suite of $CO_2$ concentration observations from various sources on the North Slope for our analysis. The United States
(US) National Oceanic and Atmospheric Administration (NOAA) Barrow Atmospheric Baseline Observatory (BRW) tall
tower near Utqiaġvik, Alaska has made continuous in situ $CO_2$ concentration measurements since 1973 (Sweeney et al., 2016).
The US Department of Energy (DOE) Atmospheric Radiation Measurement Climate Research Facility Airborne Carbon
Measurements V (ARM-ACME V) airborne campaign measured $CO_2$ concentrations sub-weekly from June to September
2015 over the North Slope (Biraud et al., 2016; Tadić et al., 2021). The US National Aeronautics and Space Administration
(NASA) Arctic-Boreal Vulnerability Experiment (ABoVE) Arctic Carbon Atmospheric Profiles (Arctic-CAP) airborne
campaign flew throughout Alaska and northwestern Canada approximately every month from May to November 2017
(Sweeney and McKain, 2019; Sweeney et al., 2022). $CO_2$ concentration observations from the NASA Carbon in Arctic
Reservoirs Vulnerability Experiment (CARVE) flights for 2012–2014 are incorporated into the Commane et al. (2017)
optimized $CO_2$ fluxes used in our analysis below. The NOAA/US Coast Guard collaborative Alaska Coast Guard (ACG)
flights have also made aircraft $CO_2$ concentration measurements in the region, but these coastal flights observe only limited
spatial coverage of the North Slope, and we do not use them here.

For the NOAA BRW tower, we use hourly $CO_2$ concentration observations with wind direction from the land (135°–

202.5° clockwise w.r.t. north) and ocean sectors (0°–45°), avoiding Utqiaġvik anthropogenic activity, with wind speed > 2.5
m s$^{-1}$ (Fig. S2) (Commane et al., 2017; Sweeney et al., 2016). We only use land sector observations from the cold season
(defined here as September–April) since seasonal wind patterns do not favor transport from those directions during the growing
season (defined here as May–August). For the ARM-ACME V and ABoVE Arctic-CAP aircraft campaign observations, we
group averaged sampling points into 50 m vertical bins after removing data influenced by combustion sources such as
anthropogenic activity and biomass burning events. These combustion sources of $CO_2$ are expected to be small (<1 TgC yr$^{-1}$
on the North Slope, see Table S1) during our study period. They are not accounted for in biogenic $CO_2$ flux models, however,
and must be removed from our analysis when observed. We remove time periods with elevated carbon monoxide (CO)
concentration above 150 ppb, as in Chang et al. (2014) and Commane et al. (2017), which indicates local combustion sources.
Time periods with highly variable CO concentrations ($\Delta CO > 40$ ppb) indicate complex mixing of more remote combustion
sources and are also removed (Chang et al., 2014). The remaining grouped sampling points correspond to the available
Lagrangian atmospheric transport modeling system simulations (WRF-STILT (Henderson et al., 2015), see below): ARM-
ACME V points are calculated every 50 m vertically below 1 km, every 100 m vertically above 1 km, and every 10 km
horizontally from 1 s observations, and ABoVE Arctic-CAP points are matched every 20 s from averaged 10 s observations.
To ensure these points observe the Alaska North Slope, we only use points with at least 70% of the total 10-day WRF-STILT
simulated surface influence occurring in our regional domain.

**2.1.2 Eddy covariance $CO_2$ flux tower observations**

We also use up to five years (2013–2017) of year-round observations of net $CO_2$ flux from eight eddy covariance tower sites
(for 32 total site-years) representing an array of tundra ecosystems throughout the Alaska North Slope (Figs. 1b, S1, Table S2
in Supplement). These half-hourly eddy flux measurements of net $CO_2$ flux are not gap-filled to avoid introducing additional
uncertainties. Three of the sites are located near Imnavait Creek along a wetness gradient from valley to hilltop: wet sedge
tundra (ICS), moist acidic tussock tundra (ICT) and dry heath tundra (ICH) (Euskirchen et al., 2017, 2012). The other sites
include tussock tundra at Ivotuk (IVO), wet polygonised tundra at Atqasuk (ATQ), and three sites near Utqiaġvik: wetland
tundra (BES), wet polygonised tundra (BEO), and moist tundra (CMDL) (Zona et al., 2016; Arndt et al., 2020).

**2.2 Observed atmospheric $CO_2$ concentration enhancement calculation**

We calculate the observed *top-down* atmospheric $CO_2$ concentration enhancement ($\Delta CO_2$) for the North Slope region for every
land-sector hour at the NOAA BRW tower and for every 50 m of vertical distance transited during the airborne campaigns
(ARM-ACME V, ABoVE Arctic-CAP). The observed $\Delta CO_2$ [units: ppm] generated by the North Slope ecosystem is
calculated relative to the background concentration without influence from this region such that:

observed $\Delta CO_2$ = observed $[CO_2]$ – background $[CO_2]$                      (1)

following previous work (Sweeney et al., 2016; Commane et al., 2017; Jeong et al., 2018).

The background $CO_2$ concentrations at the NOAA BRW tower are determined by smoothing the 10-day mean of the

observed ocean sector concentrations using spline fitting to produce a daily $CO_2$ background concentration. We calculate the
uncertainty of these background concentrations by both 1) varying the starting hour of the 10-day mean calculation prior to
spline fitting and 2) randomly sub-selecting 50% the ocean sector concentrations 1000 times. The interval that contains 95%
of these 240,000 fits represents our daily background uncertainty. Figure S2 shows the ocean sector concentrations, resulting
background concentration, and uncertainty described here.
To determine the background $CO_2$ concentrations for the ARM-ACME V and ABoVE Arctic-CAP aircraft
campaigns, we isolate aircraft observations without surface influence from the North Slope using the WRF-STILT footprints
as done for larger regions in Chang et al. (2014) and Commane et al. (2017). These observed $CO_2$ concentrations represent the
state of the air before it interacts with the surface in the study region. The regional backgrounds vary by the direction from
which the air enters the domain. For example, the backgrounds from the south and from over land generally experience $CO_2$
drawdown prior to those from over the Arctic Ocean. The time- and directional-dependent backgrounds we use are shown in
Fig. S3. We apply the uncertainty from the NOAA BRW tower background to the aircraft backgrounds as a reasonable
representation of the variability associated with available background $CO_2$ concentration data.
**2.3 Simulated atmospheric $CO_2$ concentration enhancement calculation**
To understand how landscape interactions with the atmosphere (through $CO_2$ flux) influenced the observed $CO_2$ concentrations
across space and time, we calculate the corresponding simulated $\Delta CO_2$ [units: ppm] by transporting *bottom-up* biogenic $CO_2$
fluxes to each observation site such that:
$$\text{simulated } \Delta CO_2 = \text{simulated } CO_2 \text{ flux} \times \text{simulated footprint} \qquad (2)$$
In this calculation, we multiply the hourly simulated $CO_2$ flux [µmol $CO_2$ m$^{-2}$ s$^{-1}$] by the footprint [ppm (µmol $CO_2$ m$^{-2}$ s$^{-1}$)$^{-}$
$^1$] for that hour starting at the observation point, backward in time for each hour up to ten days, where the footprint quantifies
the influence of the land surface on the concentration observed at a measurement point. The simulated $\Delta CO_2$ is then the sum
of these hours.
We use expected $CO_2$ fluxes based on a variety of bottom-up model approaches which represent North Slope
ecosystems. Year-round bottom-up estimates of net $CO_2$ fluxes (defined by the models as net ecosystem exchange, NEE) are
obtained from the Tundra Vegetation Photosynthesis and Respiration Model (TVPRM) ensemble, and from existing model
output from Luus et al. (2017) and Commane et al. (2017). Independent bottom-up estimates of belowground $CO_2$ emissions
(= NEE) for the cold season (net $CO_2$ uptake = 0) were obtained from Natali and Watts et al. (2019) and Watts et al. (2021).
The TVPRM model ensemble development process is described in Sect. 2.4, and the other $CO_2$ flux models, including their
native spatial and temporal resolutions, are listed in Table S3.
The footprints are generated by the Lagrangian atmospheric transport modeling system, WRF-STILT (Stochastic
Time-Inverted Lagrangian Transport model driven by Weather Research and Forecasting model meteorology (Henderson et
al., 2015)). In this system, WRF meteorological fields are first generated for the study region and time period (v3.5.1 for ARM-
ACME V and NOAA BRW tower footprints used here, v3.9.1 for ABoVE Arctic-CAP footprints). STILT then uses the WRF
meteorology to estimate the contribution of surface fluxes to the atmospheric concentration at a specified time and place, called
a receptor, by calculating the amount of time air (represented by a distribution of particles) spends in the lower half of the
boundary layer at a given location. The WRF-STILT model configurations from Henderson et al. (2015) have been used
extensively in numerous previous papers to study greenhouse gas fluxes using observations from aircraft and towers in Alaska,
including on the North Slope (e.g., Chang et al., 2014; Miller et al., 2016; Zona et al., 2016; Commane et al., 2017; Karion et
al., 2015; Hartery et al., 2018). An evaluation by Henderson et al. (2015) for WRF v.3.4.1 and v3.5.1 showed that their polar
WRF configuration performs well against surface observations of air temperature and wind speed in Alaska and that WRF-
STILT can capture the shape and approximate depth of greenhouse gases in the column. Zona et al. (2016) note that WRF
planetary boundary layer ventilation rates may be biased in the fall (and winter) when heat fluxes are low, but this error is
difficult to assess quantitatively. For this study, we use receptors set to correspond with the tower and aircraft $CO_2$
concentration observations. The footprints (and their corresponding measurements) for these receptors sample air from
throughout the North Slope but are concentrated more heavily toward the area around the NOAA BRW tower (Fig. 1c).

For calculating simulated $\Delta CO_2$ from the TVPRM ensemble, we grid the distribution of WRF-STILT particles and
their corresponding surface influence to the spatial resolution of the meteorological reanalysis products driving the model. The
$CO_2$ flux models used for comparison to the TVPRM ensemble are similarly treated using 0.5°-gridded 10-day WRF-STILT
footprints, which are available on a circumpolar grid poleward of 30°N. The simulated $CO_2$ fluxes from Luus et al. (2017),
Natali and Watts et al. (2019), and Watts et al. (2021) are regridded to 0.5° spatial resolution. For the models by Natali and
Watts et al. (2019) and Watts et al. (2021), which only estimate monthly $CO_2$ fluxes, we apply a constant flux for that month.
Since the ends of our defined cold season (September–April) include transitional periods when some biogenic plant activity
does occur (hence belowground $CO_2$ emissions $\neq$ NEE), for the Natali and Watts et al. (2019) and Watts et al. (2021) bottom-
up scenarios, we add in estimates of photosynthesis and plant respiration fluxes from the TVPRM ensemble for April and
September.
**2.4 Empirically simulated biogenic $CO_2$ fluxes from tundra ecosystems**
We develop the TVPRM as an ensemble of ecosystem-resolved models that represent a more extensive range of potential
tundra ecosystem functional relationships, environmental drivers, and scaling assumptions than available from other $CO_2$ flux
models. For this study, TVPRM generates a set of spatially and temporally varying $CO_2$ flux maps for a six-year period (2012–
2017) at $30 \times 30$ km spatial and 1 hr temporal resolution for the Alaska North Slope.

TVPRM is driven by parameterized functional relationships for soil respiration ($R_{soil}$), plant respiration ($R_{plant}$), and
photosynthesis (gross primary productivity (GPP)), which are described by:
$$R_{soil} = \alpha_s \times T_s + \beta_s \tag{3}$$
$$R_{plant} = \alpha_a \times T_a + \beta_a \tag{4}$$
$$GPP = \lambda \times T_{scale} \times SIF \times PAR \times \frac{1}{1 + \frac{PAR}{PAR_0}} \tag{5}$$
$$T_{scale} = \frac{(T_a - T_{min})(T_a - T_{max})}{(T_a - T_{min})(T_a - T_{max}) - (T_a - T_{opt})^2} \tag{6}$$
The simulated hourly $CO_2$ fluxes [units: µmol $CO_2$ m$^{-2}$ s$^{-1}$] are determined as responses to light and heat: $R_{soil}$ is a function of
near-surface soil temperature ($T_s$) [°C]; $R_{plant}$ is a function of air temperature ($T_a$) [°C]; and GPP is a function of a temperature
scalar ($T_{scale}$) and photosynthetically active radiation (PAR) [µmol photon m$^{-2}$ s$^{-1}$], with solar-induced chlorophyll fluorescence
(SIF) [mW m$^{-2}$ nm$^{-1}$ sr$^{-1}$] used to define the seasonal cycle of photosynthetic capacity. $T_s$ depths are determined by reanalysis
product and listed in Table S4. $T_{scale}$ ranges from 0 to 1 based on the position of $T_a$ on the continuum between minimum
temperature ($T_{min} = 0°C$), maximum temperature ($T_{max} = 40°C$), and optimal temperature ($T_{opt} = 15°C$). NEE is then calculated
as:
$$NEE = R_{soil} + R_{plant} - GPP \qquad\qquad (7)$$
with positive NEE values indicating a net source of $CO_2$ into the atmosphere and negative NEE values meaning net movement
of $CO_2$ into the biosphere. We use NEE to be synonymous with net $CO_2$ flux. Using SIF, which correlates to photosynthetic
activity (Porcar-Castell et al., 2014; Yang et al., 2015), in the modeling framework provides an advantage over indices such
as enhanced vegetation index (EVI) due to the limited canopy and evergreen nature of tundra ecosystems (Luus et al., 2017).
The parameter values ($\alpha_s$, $\beta_s$, $\alpha_a$, $\beta_a$, $\lambda$, $PAR_0$) for the site-level relationships used by TVPRM are determined first
using the observed net $CO_2$ fluxes from the eddy flux sites (see Sect. S1 in Supplement). We determine the site-level parameters
separately for each combination of reanalysis product (NARR (Mesinger et al., 2006) and ERA5 (Hersbach et al., 2020)),
which provide $T_a$, $T_s$, and PAR, and SIF product (GOME-2 (Joiner et al., 2016), GOSIF (Li and Xiao, 2019), and CSIF (Zhang
et al., 2018)) that will later be used to generate the regional TPVRM ensemble (Tables S4–S5, see Sects. S2–S3). Additional
$\alpha_s$ and $\beta_s$ parameters are determined using $T_s$ from the Remote Sensing driven Permafrost Model (RS-PM (Yi et al., 2019,
2018)) to test its implementation in TPVRM. RS-PM uses tailored input for Alaska permafrost zones, such as downscaled
snow depth and aircraft-observed soil dielectric constants and was developed and tested using $T_s$ and active layer thickness
measurements from the North Slope. RS-PM also produces $T_s$ at higher vertical resolution in the near-surface than the
reanalysis products to capture subsurface heterogeneity in unfrozen soil, which is important to represent the zero-curtain
throughout the freezing and thawing periods in Alaska.
Using the median parameter value sets for each site, we simulate the TVPRM net $CO_2$ flux for our study period at
every site location to perform a cross-site evaluation (Fig. S1). These simulated net $CO_2$ fluxes perform well against the net
$CO_2$ flux observations at their corresponding sites (Figs. 1d, S4, see Sect. S4). This process also identifies two distinct
ecosystem groups: "inland", predominately graminoid and shrub tundra (ICS, ICT, ICH, IVO), and "coastal", predominately
wetland tundra (ATQ, BES, BEO, CMDL), based on the similar simulated $CO_2$ flux responses to the meteorology- and SIF-
determined functional relationships within each group demonstrated by the cross-site evaluation (Fig. S1).
The net $CO_2$ flux for each meteorological grid box in our study domain is then calculated using the site-level
functional relationships for both tundra groups. These fluxes are weighted by the spatial distribution of inland and coastal
tundra from three different vegetation maps (CAVM (Walker et al., 2005), RasterCAVM (Raynolds et al., 2019), and ABoVE
LC (Wang et al., 2020), Fig. S5, Table S6, see Sect. S5) to produce the regionally scaled TVPRM net $CO_2$ flux. By varying
the choice of representative inland and coastal tundra sites, meteorological reanalysis product, vegetation map, and SIF
product, we generate 288 different simulations (members) of net $CO_2$ flux (referred to here as the unconstrained TVPRM
ensemble) for each grid box across the region for each of the six study years. Monthly and annual regional net $CO_2$ flux values
are calculated as the area-weighted sum of all grid boxes simulated in our domain. Notable changes since the previous iteration
of this empirical $CO_2$ flux model (Commane et al., 2017; Luus et al., 2017) include the expansion of the model to include
multiple ensemble members to account for variability and uncertainty in model formulation, the use of additional site-years of
$CO_2$ flux observations (with increased data coverage over the cold season), more inclusive data filtering methods, and much
higher temporal (1-, 4-, and 8-day rather than monthly) and spatial (0.01° and 0.05° rather than 0.5°) resolution SIF datasets.
We compare TVPRM to the previous model version by Luus et al. (2017) and its CARVE-informed optimization by Commane
et al. (2017) in Sect. 3.3.

## 2.5 Evaluation Framework

We use the atmospheric $CO_2$ concentration observations to evaluate the many tundra ecosystem parameterizations, vegetation
distributions, and environmental drivers that represent the net $CO_2$ flux on the North Slope over various spatial and temporal
scales. For this assessment, we compare the observed $\Delta CO_2$, which are the observed $CO_2$ concentration changes driven by
regional $CO_2$ fluxes, with the simulated $\Delta CO_2$ determined by combining the regional biogenic $CO_2$ flux models with the
atmospheric transport model.
To compare the regional observed $\Delta CO_2$ and simulated $\Delta CO_2$, we calculated the coefficient of determination ($R^2$) as
the square of the Pearson correlation coefficient for all points. The slope (m) is determined by ordinary least squares using the
median of each 10% bin of ordered observed and corresponding simulated net $CO_2$ flux. The normalized mean bias (NMB) of
all points is defined as $\frac{\sum (\text{simulated} - \text{observed})}{\sum \text{observed}}$ . The root-mean-square error (RMSE) of all points is defined as
$\sqrt{(\text{simulated} - \text{observed})^2}$.
These comparisons enable us to constrain the regional net $CO_2$ flux on the Alaska North Slope. First, we identify the
year-round empirically driven net $CO_2$ fluxes from the TVPRM ensemble (TVPRM Unconstrained) which are most consistent
with the $CO_2$ concentration observations from the two aircraft campaigns and at the tower (TVPRM Constrained) (Sects. 3.1–
3.2). Then, noting the large range in potential cold season $CO_2$ fluxes, we compare our constrained TVPRM member with $CO_2$
fluxes from previous studies (Sect. 3.3). Finally, we suggest and quantify sources of the missing $CO_2$ flux observed during the
early cold season (defined here as September–December) and incorporate those fluxes into our net $CO_2$ budget (TVPRM
Constrained + Additional Zero Curtain Emissions (ZC) and Inland Water Fluxes (IW)) (Sect. 3.4). This analysis provides a
unique regional net $CO_2$ flux quantification for the North Slope that is verified using atmospheric observations and can also
be explained from an ecological and physical perspective.

## 3. Results

### 3.1 Evaluation of unconstrained empirical net $CO_2$ flux model ensemble

#### 3.1.1 Using aircraft-observed $CO_2$ enhancements

The observed $\Delta CO_2$ during the ARM-ACME V (June–September 2015) and ABoVE Arctic-CAP (May–November 2017) airborne campaigns show a strong seasonal uptake pattern throughout the growing season (Figs. 2a–2b). The frequent flights during ARM-ACME V (multiple flights per week) observe the transition from early to peak growing season uptake (observed $\Delta CO_2 = -11$ ppm) and on into cold season respiration, which results in net $CO_2$ source conditions in September (+5 ppm). While less frequent, the ABoVE Arctic-CAP flights begin at the end of the cold season, extend later into following cold season, and cover a larger area of the North Slope. Peak growing season uptake observed by the ABoVE Arctic-CAP flights (–14 ppm) is slightly stronger than for during ARM-ACME V, and by November, the ABoVE Arctic-CAP flights observe a strong $CO_2$ source throughout the North Slope (+10 ppm). The difference in observed $\Delta CO_2$ during peak growing season uptake between 2015 and 2017 is likely similar to the uncertainty in the respective values and could be due to differences in areas of the North Slope sampled between years.

The magnitude and timing of the observed net $CO_2$ uptake throughout the growing season is generally well represented by the empirical net $CO_2$ flux model ensemble (TVPRM Unconstrained, Figs. 2a–2b, S6). The median coefficients of determination ($R^2$) and ordinary least squares slopes between the observed and simulated $\Delta CO_2$ for this time are 0.54 and 0.41 for ARM-ACME V and 0.82 and 0.72 for ABoVE Arctic-CAP, respectively. Only for the July observations during the ABoVE Arctic-CAP campaign do many members of the $CO_2$ flux trend toward an underestimate of net $CO_2$ uptake, with all points showing a much larger range in simulated values compared to ARM-ACME V. The net $CO_2$ release tends to be overestimated by the TVPRM ensemble during the ABoVE Arctic-CAP seasonal transitions in May and September, but during November the observed $R_{soil}$ is consistently underestimated.

Given the large range of unconstrained representations of the regional $CO_2$ flux, the accuracy in simulating the aircraft observed $\Delta CO_2$ varies between TVPRM ensemble members. For example, members using the RasterCAVM vegetation map, which places less coastal tundra area cover in the south (Fig. S5), produce a smaller mean July net $CO_2$ uptake flux (by ~1 µmol m$^{-2}$ s$^{-1}$, Fig. S7a) throughout the southern North Slope than members using other vegetation maps (CAVM and ABoVE LC), and this placement consistently underestimates the net $\Delta CO_2$ uptake during the growing season compared to the aircraft observations by 5–10 ppm (Fig. S8). Also, members driven by SIF products that integrate additional remote sensing and/or meteorological data (GOSIF and CSIF) better reflect the timing and magnitude of the peak season carbon uptake in tundra ecosystems than members produced by interpolated SIF retrievals (GOME-2 SIF product), which underestimate the observed $CO_2$ uptake during July (Fig. S8).

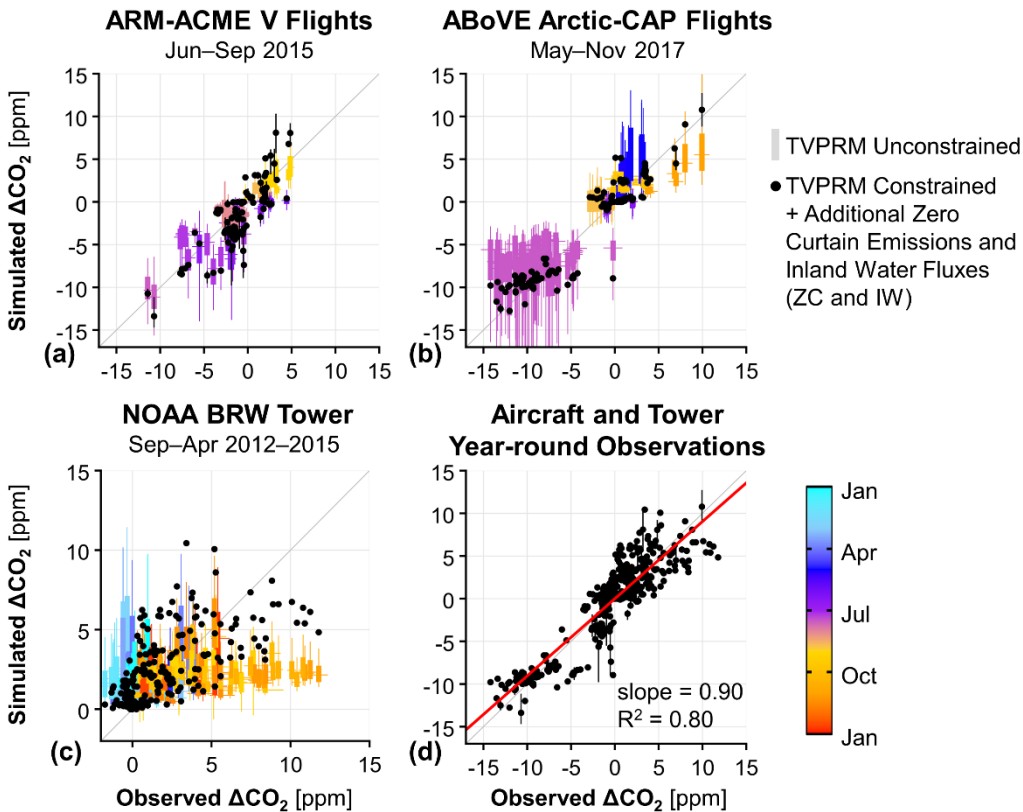

**Figure 2.** Aircraft and tower $CO_2$ concentration measurements constrain year-round simulated $CO_2$ fluxes on the Alaska North Slope. **(a)**–
**(c)** Comparison of observed and simulated $\Delta CO_2$ during the ARM-ACME V flight campaign **(a)**, during the ABoVE Arctic-CAP flight
campaign **(b)**, and at the NOAA BRW tower **(c)** for air over the Alaska North Slope. Horizontal lines indicate range of uncertainty in the
NOAA BRW tower ocean sector background calculation. Vertical boxes colored by month of the year represent 50% and whiskers represent
95% of $\Delta CO_2$ values from all members of unconstrained TVPRM ensemble (see Sect. 2.4) from all binned points. Black points show values
from the constrained TVPRM member with additional zero-curtain emissions (ZC) and inland water fluxes (IW) (see Sect. 3.4). For **(a)**–**(b)**,
observed values are vertically binned medians, and for constrained TVPRM member + ZC and IW, vertical lines contain middle 95% of
$\Delta CO_2$ values from all binned points. **(d)** Combined comparison of observed and simulated $\Delta CO_2$ for all aircraft and tower points using
constrained TVPRM member + ZC and IW. Shown with linear best fit (red line), slope determined by ordinary least squares, and coefficient
of determination ($R^2$) of all points (n = 455). 1:1 line shown in dark gray.

Using these comparisons, we identify less-representative ensemble members that generally underestimate the
observed $\Delta CO_2$ uptake during the growing season (RasterCAVM vegetation map and GOME-2 SIF product members).
Removing these members from the TVPRM ensemble improves the collective performance of the remaining members during
the growing season (Fig. S6), brings the median slope of agreement closer to 1 for both campaigns (improves from 0.53 to
0.64 and from 0.71 to 0.94 for ARM-ACME V and ABoVE Arctic-CAP, respectively), and reduces median NMB (–0.34 to –
0.03) and median RMSE (3.12 to 2.73) for ABoVE Arctic-CAP.

### 3.1.2 Using tower-observed CO₂ enhancements

As seen with the September–November aircraft data, the observed $\Delta CO_2$ at the NOAA BRW tower (Fig. 2c) indicate that the $CO_2$ source to the atmosphere increases substantially from September to peak in October and November (+12 ppm) before decreasing to near zero throughout the late cold season (January–April).

Most of the TVPRM ensemble members substantially underestimate the observed $\Delta CO_2$ in the early cold season (September–December) as the soils freeze, and some simulations produce too much $CO_2$ in the late cold season when the soils are frozen (Fig. 2c). The cold season $CO_2$ flux differs greatest in magnitude and spatial extent between the ensemble members parameterized for the ICS and ICT inland tundra sites (Figs. 3a, S9–S10), with a net $CO_2$ flux difference of ~0.2 µmol m$^{-2}$ s$^{-1}$ throughout the region (Fig. S7b).

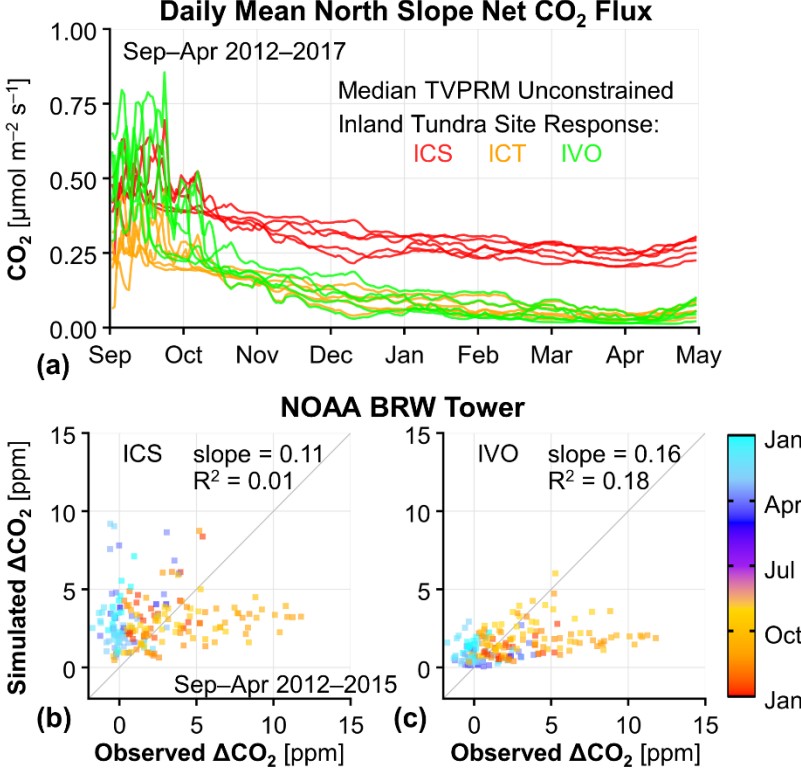

**(a)**

**(b)** **(c)**

**Figure 3.** Cold season $CO_2$ emissions for inland tundra site parameterizations and comparison to tower observations. **(a)** Timeseries of simulated daily mean Alaska North Slope net $CO_2$ flux for the median of all unconstrained TVPRM ensemble members using each of three inland tundra site parameterizations: ICS (red), ICT (orange), and IVO (green). Yearly colored lines shown for Sep–Apr beginning in Sep 2012 and ending in Apr 2017. Same for all eight eddy flux sites shown in Fig. S9. **(b)–(c)** Comparison of observed and simulated $\Delta CO_2$ at the NOAA BRW tower for air over the North Slope using the median of all unconstrained TVPRM ensemble members for the inland tundra site parameterizations at ICS (b) and IVO (c). All points colored by day of year. Shown with slope determined by ordinary least squares and coefficient of determination ($R^2$) of all points (n = 191). 1:1 line shown in dark gray.

While the magnitude of $CO_2$ flux from ICS members better matches the observed $\Delta CO_2$ in the early cold season than from other sites (Figs. 3b–3c, S11), the response to $T_s$ at ICS shows only a modest decrease in $CO_2$ flux between the early and

late cold season (Fig. 3a, 32% decrease between October and March), resulting in an overestimate of the regional $\Delta CO_2$ in the
late cold season. The $CO_2$ flux response to $T_s$ for ICT members is similar to that for ICS but lower in magnitude, and the
simulated $\Delta CO_2$ from members of neither site performs well against the observations in both the early and late cold season.
Therefore, ICS and ICT inland tundra responses to $T_s$ are not representative of the regional $\Delta CO_2$ observed at the NOAA BRW
tower throughout the entire cold season, and we remove those members from our TPVRM ensemble.

The observed net $CO_2$ fluxes at the IVO inland tundra and CMDL coastal tundra sites both show prolonged zero-

curtain emissions (Fig. S1) and respond strongly to $T_s$ in the early cold season (Fig. S9). The stronger response of $CO_2$ fluxes
to $T_s$ from the early to late cold season at IVO (Fig. 3a, 70% decrease by January) compared to at the Imnavait Creek sites
produces TVPRM members that better represents the large regional decrease in $\Delta CO_2$ observed on the North Slope (Fig. 3c).
While all coastal tundra sites respond similarly to $T_s$ during the cold season, we determine that the $CO_2$ flux magnitude at
CMDL is most consistent with the regional observations (Fig. S11). $T_s$ from ERA5 remain warmer throughout the late cold
season compared to those from NARR, which causes simulations using ERA5 $T_s$ to overestimate $CO_2$ release during that time
(Fig. S11). Unlike during the growing season, cold season $CO_2$ fluxes are not sensitive to the vegetation distribution and SIF
products.

Finally, we identify the TVPRM member that best matches the observed $\Delta CO_2$: parameterized by IVO inland tundra

and CMDL coastal tundra site responses, distributed by the ABoVE LC vegetation map, and driven by NARR reanalysis and
the CSIF SIF product (referred to here as TVPRM Constrained, Figs. S6, S12). This constrained simulation estimates a mean
regional $CO_2$ flux of 0.05 µmol m$^{-2}$ s$^{-1}$ for the late cold season in 2012–2015 and reproduces well the observed $\Delta CO_2$ during
this time (Fig. 4a). The late cold season NMB and RMSE against the observations at the NOAA BRW tower are reduced from
4.91 to 2.04 and from 1.94 to 1.30, respectively, for the constrained simulation compared to the median of the entire TVPRM
ensemble (Fig. S12). However, the early cold season $CO_2$ emissions, with a mean regional $CO_2$ flux of 0.25 µmol m$^{-2}$ s$^{-1}$ for
September–December (Fig. S13a), are still underestimated, with the simulated $\Delta CO_2$ lower than the observed $\Delta CO_2$ by ~5
ppm (Fig. 4a).
**3.2 Alternative $T_s$ products and $R_{soil}$ parameterizations**
To test the impact of reanalysis $T_s$ on the early cold season $CO_2$ fluxes, we implement $T_s$ that are more specifically developed
to represent Alaska tundra permafrost soils during freeze-thaw processes than the reanalysis products driving our constrained
TPVRM member. A single layer of $T_s$ at 8 cm depth from RS-PM (Fig. S14a) captures the magnitude and temporal behavior
of the observed early cold season $CO_2$ fluxes slightly better than the constrained member (Figs. 4a, S12), which uses NARR
reanalysis $T_s$ and does not incorporate permafrost-model derived $T_s$. The RS-PM $T_s$ extends $CO_2$ emission fluxes further into
the cold season by up to a month, which is consistent with a better representation of the zero-curtain period, however, emissions
remain higher throughout the late cold season than our atmospheric observation-constrained $CO_2$ fluxes (Fig. S15). We also
test the implementation of a multi-layer fit driven by soil column temperature from RS-PM, but neither of these instances of
remote sensing informed $T_s$ substantially improve the agreement of the $\Delta CO_2$ at the NOAA BRW tower during the early cold
season. Attempts to use alternative $R_{soil}$ formulations based on $T_s$, including $Q_{10}$ relationships, also fail to reproduce the
observed elevated $CO_2$ fluxes during the cold season.

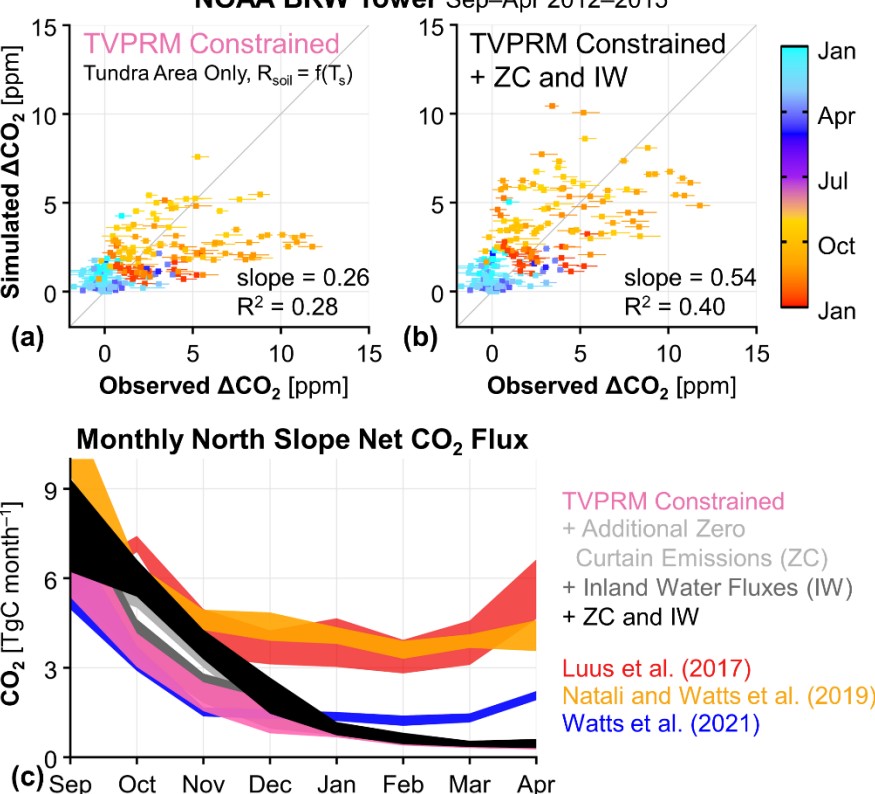


**Figure 4.** Tall tower atmospheric observations of the Alaska North Slope support early cold season emissions not driven by soil temperature ($T_s$) and present no evidence for elevated late cold season emissions. **(a)**–**(b)** Comparison of hourly cold season (Sep–Apr) observed and simulated $\Delta CO_2$ at the NOAA BRW tower for the constrained TPVRM member, where soil respiration ($R_{soil}$) is determined only by $T_s$ **(a)** and for the constrained TVPRM member + additional zero-curtain emissions (ZC) and inland water fluxes (IW) **(b)**. Horizontal segments indicate range of uncertainty in the NOAA BRW tower ocean sector background calculation. Shown with slope determined by ordinary least squares and coefficient of determination ($R^2$) of all points (n = 191). 1:1 line shown in dark gray. **(c)** Monthly cold season total Alaska North Slope net $CO_2$ fluxes for various $CO_2$ flux models. TVPRM-based simulations and Natali and Watts et al. (2019) show values for 2012–2017, Luus et al. (2017) show 2012–2014, and Watts et al. (2021) show Sep 2016–Apr 2017. Ribbons represent range of all years, where applicable. Area of the North Slope domain used to calculate regional totals is $3.537\times10^5$ km$^2$.

**3.3 Evaluation of other CO₂ flux models during the cold season**

More early cold season (September–December) $CO_2$ flux into the atmosphere is observed at the NOAA BRW tower than is
emitted by our constrained empirical simulation member, and these observations also indicate low late cold season (January–
April) $CO_2$ emissions. We compare our constrained $CO_2$ fluxes to several other representations of gridded $CO_2$ flux on the
North Slope (Table S3) and find that difficulty in simulating the magnitude and timing of the observed $\Delta CO_2$ throughout the
cold season is not unique to the constrained fluxes from our study.
The net $CO_2$ fluxes from Luus et al. (2017) are similar to the constrained TVPRM member during the growing season
(Fig. S16), but release more than three times as much $CO_2$ into the atmosphere throughout the late cold season (Fig. 4c). This
large late cold season $CO_2$ flux leads to a large overestimate compared to the observed $\Delta CO_2$ (Fig. S14b). The optimization
employed by Commane et al. (2017) increases the September–October $CO_2$ flux to a range that matches our observations at
the NOAA BRW tower. However, Commane et al. (2017) did not optimize the cold season fluxes from November to March,
but reverted to Luus et al. (2017) fluxes during this time, thus producing late cold season fluxes that are too large. Overall,
Commane et al. (2017) projected a regional total cold season $CO_2$ source of 37–40 TgC for 2012–2014, which is more than
twice as high as our constrained TVPRM member $CO_2$ flux (15–18 TgC) for those years.
Carbon dioxide fluxes from work by Natali and Watts et al. (2019), a cold season model developed for the global
high latitude permafrost region, are similar to our constrained TVPRM member in September, but the fluxes remain high
throughout the cold season (Fig. 4c) similarly to Luus et al. (2017), for a range of total cold season $CO_2$ flux of 40–43 TgC for
2012–2017. This sustained $CO_2$ release also leads to an overestimation in the $\Delta CO_2$ in the late cold season for this region (Fig.
S14c). Tao et al. (2021) also show that the cold season $CO_2$ fluxes of Natali and Watts et al. (2019) are high compared to their
model. More recent work by Watts et al. (2021), using observations from new Soil Respiration Station monitoring sites in
Alaska, produces cold season $CO_2$ fluxes more similar to our constrained $CO_2$ fluxes, with an underestimate in the simulated
$\Delta CO_2$ during the early cold season (Fig. S14d), for a total cold season $CO_2$ flux of 18 TgC for September 2016 to April 2017.
**3.4 Sources of missing $CO_2$ fluxes**
None of the flux products discussed above, including our TVPRM ensemble, account for any potential $CO_2$ fluxes during the
zero-curtain period that are not driven by $T_s$ or are from areas on the terrestrial-aquatic interface. To account for these processes,
we first add an additional $CO_2$ flux with zero-curtain timing to our constrained $CO_2$ flux (TVPRM) member from both inland
and coastal tundra areas that consists of 0.25 µmol $m^{-2}$ $s^{-1}$ for October with a reduction to zero by the end of December. This
peak additional $CO_2$ flux is within the daily variability of the observed $CO_2$ flux at the IVO and CMDL eddy flux sites during
the zero-curtain period (Fig. S9) and the reduction into December is consistent with these observations. The additional zero-
curtain flux improves the ability of the model to reproduce the observed $\Delta CO_2$ at the NOAA BRW tower (slope = 0.46, $R^2$ =
0.41). We also apply the coastal tundra site ecosystem parameterization used in our constrained TVPRM member to all areas
of inland water on the North Slope, which account for 4% of the domain according to the ABoVE LC map (Fig. S5) and were
previously set to zero $CO_2$ flux. Representing these aquatic areas with biogenic $CO_2$ fluxes consistent with coastal tundra
ecosystems is one simple way to bridge the terrestrial-aquatic gap in tundra ecosystem models, where portions of aquatic
systems on the land-water gradient (i.e., the edges) may be more likely to respond to the environment as coastal tundra than
with the zero-flux assumed by water area. The ice phenology for areas of inland water producing $CO_2$ flux is then considered
to be similar to that of the freeze-thaw timing in coastal tundra soils. Adding these coastal tundra fluxes to inland water areas
also improves the performance of our model (slope = 0.32, $R^2$ = 0.29 against NOAA BRW tower observations). The magnitude
of additional zero-curtain flux suggested here and the portion of inland water represented with coastal tundra site
parameterizations produce the best statistical comparison for a range of choices tested (Fig. S17).

Together, adding these zero-curtain (ZC) and inland water (IW) $CO_2$ fluxes to our constrained simulation (referred to

as TVPRM Constrained + ZC and IW) increases the mean regional $CO_2$ flux in early cold season by 70% (0.18 µmol m$^{-2}$ s$^{-1}$,
Fig. S13b) and results in a large improvement to our comparison of $\Delta CO_2$ at the NOAA BRW tower (slope = 0.54, $R^2$ = 0.40,
Figs. 4b, S12) and across the region using airborne data, especially during the November ABoVE Arctic-CAP flights (Figs. 2,
S6). The year-round comparison using all available aircraft and tower observations shows these net $CO_2$ fluxes are now
representative of the region (slope = 0.90, $R^2$ = 0.80, Fig. 2d). As a result, the North Slope regional total cold season $CO_2$ flux
increases by 6 TgC (~38%) to 20–24 TgC for 2012–2017 compared to the constrained empirical $CO_2$ flux model member.

### 3.5 Alaska North Slope annual net $CO_2$ flux

The median Alaska North Slope annual net $CO_2$ flux from the TVPRM ensemble (–5 TgC yr$^{-1}$) for 2012–2017 is consistent
with the previous multi-model comparison (Fisher et al., 2014), but we find a much smaller range in regional $CO_2$ flux values
(26 TgC yr$^{-1}$ to –29 TgC yr$^{-1}$ for 95% of TVPRM members) (Fig. S18). The largest contribution to this ensemble range comes
from the difference in parameterizations determined for the ICS and ICT inland tundra sites, with TVPRM members using ICS
trending toward a net $CO_2$ source, while ICT trends toward net $CO_2$ uptake. The distribution of inland and coastal tundra
throughout the region represented by the vegetation maps also has a noticeable impact on the sign of the net $CO_2$ flux, with
members using the RasterCAVM more likely to release net $CO_2$ into the atmosphere than members using the other maps.
There is also little interannual variability in the unconstrained TVPRM ensemble, with only 2014 moving toward a net $CO_2$
source, consistent with Tao et al. (2021) for these years.

Our best quantification of the annual net $CO_2$ flux for the North Slope informed by atmospheric observations, TVPRM

Constrained + ZC and IW, indicates that the region is a small net sink for 2013 (–5 TgC yr$^{-1}$) and 2015 (–6 TgC yr$^{-1}$) and a
small net source for 2012 (+6 TgC yr$^{-1}$), 2014 (+6 TgC yr$^{-1}$), 2016 (+2 TgC yr$^{-1}$), and 2017 (+2 TgC yr$^{-1}$) (Fig. 5a). We
estimate a 10% uncertainty in the net annual $CO_2$ flux based on the slope from our final comparison with the year-round
observations (Fig. 2d). The year-round net $CO_2$ fluxes from Luus et al. (2017) (driven with NARR meteorology, monthly
GOME-2 SIF, and CAVM vegetation map) indicate the North Slope to be a strong annual net $CO_2$ source for 2012–2014 (+9
TgC yr$^{-1}$ to +15 TgC yr$^{-1}$, Fig. S18) and are inconsistent with our results. Our results are more consistent with Tao et al. (2021),
but we find a smaller range in the magnitude of net $CO_2$ flux over the same years and more years trending toward a net $CO_2$
source.

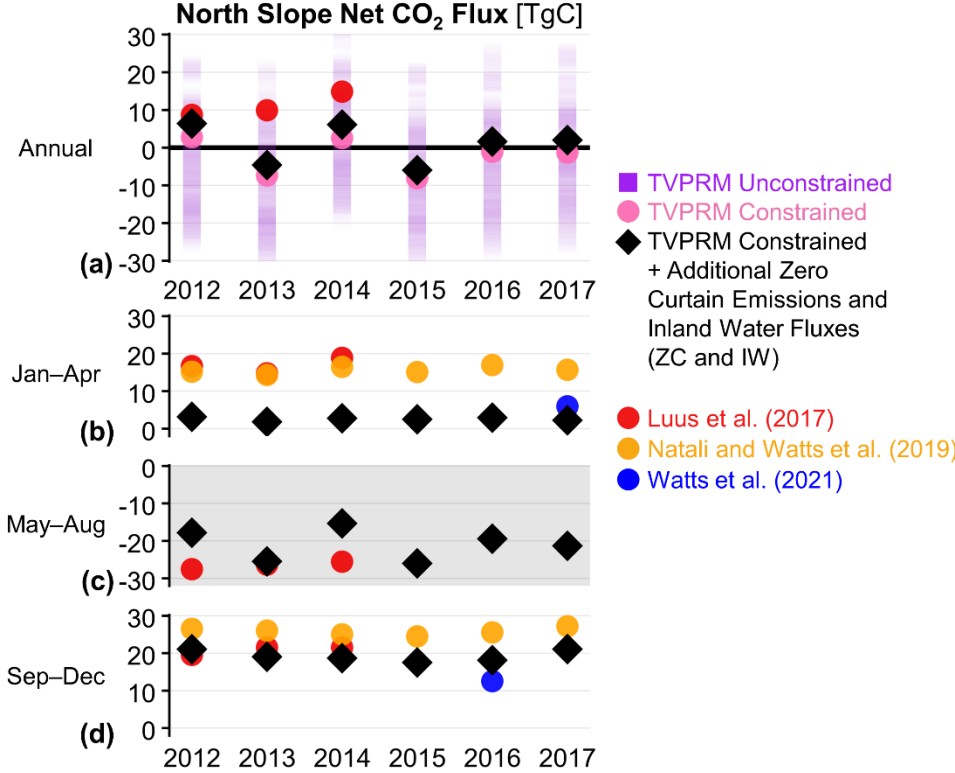

**Figure 5.** Annual and seasonal Alaska North Slope net $CO_2$ flux constrained by aircraft and tower observations. **(a)** Annual, **(b)** late cold season (Jan–Apr), **(c)** growing season (May–Aug), and **(d)** early cold season (Sep–Dec) total Alaska North Slope net $CO_2$ fluxes for various $CO_2$ flux models for 2012–2017 as in Fig. 4. Purple squares indicate middle 95% of all TVPRM ensemble members.

We find that the regional net growing season $CO_2$ uptake and the cold season emissions on the North Slope are comparable in magnitude, so the net balance could depend on small perturbations in either flux. However, the regional cold season $CO_2$ emissions for these years were relatively similar from year to year: 18–21 TgC for the early cold season (Fig. 5d), diminishing to only 2–3 TgC for the late cold season (Fig. 5b). Therefore, the interannual variability of the regional carbon balance is largely driven by fluctuating net growing season $CO_2$ fluxes during these years: greater net growing season uptake in 2013 and 2015 than in 2012, 2014, 2016, and 2017 (Fig. 5c).

## 4. Discussion

### 4.1 Tundra ecosystem growing season net $CO_2$ fluxes

The good performance of the TVPRM ensemble against the atmospheric observations during the growing season indicates that the tundra ecosystems of the Alaska North Slope respond to light and heat as quantified by PAR, $T_s$, and $T_a$, and that the net $CO_2$ flux is largely controlled by the simple $R_{soil}$, $R_{plant}$, and GPP relationships in the empirical model over this time.

The growing season of each year determines the sign of the regional annual net $CO_2$ flux during our study period, with 2013 and 2015 being strong net sinks and 2014 being the strongest net source. The relative magnitude of each component of the net $CO_2$ flux during the growing season (i.e., $R_{soil}$, $R_{plant}$, GPP) varies from year-to-year (Table S7) and helps explain the interannual variability in the net source or sink status of the North Slope. Growing season 2015 was very warm, dry, and sunny in Alaska and resulted in extreme biomass burning activity outside of the North Slope (Table S1). High regional mean $T_a$ and PAR (Table S8) and low accumulated precipitation (Table S9) in NARR confirm this was the case for North Slope as well, with high $T_a$ and PAR contributing to a very high GPP. The growing season SIF signal from the CSIF product, which determines the seasonal cycle and relative magnitude of photosynthetic activity, is also large in 2015 (Table S8), further enhancing GPP. This year and others with a larger GPP component of NEE correspond to growing seasons with stronger SIF signals, which is an indicator of increased productivity and consistent with previous studies (e.g., Magney et al., 2019; Sun et al., 2017). While fairly high $T_a$ and $T_s$ in 2015 also result in high $R_{soil}$ and $R_{plant}$, respectively, this elevated respiration is not enough to offset the very high GPP and results in a large net $CO_2$ sink. In contrast, the summer of 2014 was cool, wet, and cloudy, and the North Slope experienced very low $T_a$, PAR, and SIF signal, producing very low GPP. Lower-than-normal $T_a$ also results in very low $R_{plant}$, but as with 2015, this is not enough to offset the extremely low uptake resulting in a large net $CO_2$ source for 2014. In 2013, the other growing season with a strong net $CO_2$ sink, moderately high GPP combines with moderately low $R_{plant}$ and very low $R_{soil}$. Extremely low $T_s$ causes this very low $R_{soil}$, which, relative to moderate $T_a$ and PAR, is likely a result of above-average lingering snowpack into May (Table S9). This lingering snowpack is perhaps surprising given that the mean snowpack for the proceeding cold season was not particularly deep. The important impact that snow cover and the timing of snowmelt has on $T_s$ and carbon response in tundra ecosystems has been recently emphasized (e.g., Kim et al., 2021), and is supported by our work, which shows that the prevalence of snow in the spring may determine the sign of the regional net $CO_2$ for an entire year.

The regional net $CO_2$ flux is highly sensitive, however, to the distribution of tundra vegetation types (upland v. coastal) throughout the North Slope during the growing season. Coastal tundra takes up more $CO_2$ for a given unit PAR compared to inland tundra, based on the relationships between observed site-level net $CO_2$ flux and PAR in this study (TVPRM parameters, Fig. S1), which could be evidence for an adaptation to lower light levels. This difference is consistent with Luus et al. (2017), who calculated greater uptake at "wetland" sites like Atqasuk and Barrow than at "graminoid tundra" sites like Ivotuk and Imnavait when all driver inputs are constant and with Mbufong et al. (2014), who also found that peak growing season net uptake for constant light is greater at Barrow than at Ivotuk. The stronger $CO_2$ uptake response of coastal tundra to light is important to consider due to the fact that the vegetation distributions assessed here with more coastal tundra to the south (CAVM (Walker et al., 2005), ABoVE LC (Wang et al., 2020)) better agree with the atmospheric observations. When considering the ability of coastal tundra to take up $CO_2$ when moved toward the south, Patankar et al. (2013) saw that tundra plants exposed to additional intense light did not respond with additional uptake. Therefore, while the ecosystem response of the southern North Slope is more consistent with coastal ecosystems, it seems possible that these areas are misclassified in either our simplified two-tundra type scheme or in the vegetation maps themselves. The large variability in net $CO_2$ flux

calculated by using the different maps supports the importance of accurate ecosystem type locations in upscaling eddy flux measurements and highlights the need for improved vegetation mapping and classification schemes in the Arctic ecology research community.

**4.2 Regional-scale cold season $CO_2$ emissions**

Observations across scales, at the in-situ eddy flux towers, the NOAA BRW tower, and from aircraft, consistently show signs of large early cold season $CO_2$ emissions from ecosystems on the Alaska North Slope. However, there is no evidence of widespread elevated emissions in this region during the late cold season, contrary to other studies (Commane et al., 2017; Natali and Watts et al., 2019). The TVPRM ensemble parameterizations using terrestrial eddy flux sites and the fluxes from other terrestrial $CO_2$ models cannot reproduce both the observed magnitude and across-season timing of these cold season $CO_2$ emissions.

The largest differences in the net $CO_2$ flux between TVPRM ensemble members result from the contrasting site conditions driving the ICS and ICT $R_{soil}$ parameterizations during the cold season. When taken separately by cold season segment, ICS members perform quite well against observations at the NOAA BRW tower for early cold season and ICT members perform well for the late cold season. The contrasting performance between site parameterizations is due to the topographic and hydrologic conditions, which are quite heterogeneous over a short distance and influence the plant communities and carbon storage, at each site. The ecosystems sampled by the ICS tower are seasonally inundated and retain a deep layer of organic soil that can be respired in greater amounts longer into the early cold season, while the well-drained hillslope at ICT does not allow for accumulation of organic matter in the same way (Euskirchen et al., 2017; Larson et al., 2021). While varying topography and soil inundation throughout the North Slope means that each of these site relationships is likely to be representative of many other locations in the region with similar conditions, the early-to-late cold season reduction in $CO_2$ fluxes at these sites is not consistent with the observed regional atmospheric trend, however, and we remove the members parameterized by them from the ensemble. Individual eddy flux site parameterizations may reproduce regional $CO_2$ fluxes for a given season, but it is important to consider their response to drivers across multiple seasons when scaling from the site-level to regional domains.

The observed cold season $CO_2$ flux pattern on the North Slope may be unique to tundra ecosystems of this region. For example, the $CO_2$ fluxes from Natali and Watts et al. (2019) and Watts et al. (2021) both incorporate measurements from the North Slope. However, Natali and Watts et al. (2019) used boosted regression trees trained on belowground respiration measurements from across the pan-Arctic tundra and boreal zones, which may not be representative for our study region. The fluxes from Watts et al. (2021) are based on respiration measurements from throughout only Alaska and northwest Canada and conform better to local conditions. The evaluation of these $CO_2$ fluxes against atmospheric $CO_2$ measurements also produces results that are more consistent with our TVPRM ensemble determined by North Slope eddy flux tower measurements.

We find that the atmospheric observations are best matched by biogenic $CO_2$ fluxes that include an additional $CO_2$ source from tundra ecosystems during the zero-curtain period that are independent from $T_s$ variability and year-round net $CO_2$ fluxes

from areas of inland water. The additional zero-curtain flux represents large-scale emission events not directly timed to microbial activity and root respiration controlled by $T_s$, but could be related to the delayed physical release of previously produced $CO_2$ from soil through the snowpack as the soil layers remain unfrozen (Bowling and Massman, 2011). The Alaska North Slope also has many water bodies distributed throughout the coastal tundra region, and the extent to which carbon cycles between small, shallow ponds and their surrounding terrestrial components is unclear (Magnússon et al., 2020). The biogenic $CO_2$ fluxes in these areas are likely driven by ecosystem-scale $CO_2$ fluxes from both coastal tundra and small ponds (Holgerson and Raymond, 2016; Tan et al., 2017) and their impact on the regional net $CO_2$ flux, via both emissions and uptake, may be significant (Elder et al., 2018; Beckebanze et al., 2022). Only by adding fluxes that match observed zero-curtain $CO_2$ emission pulses and by approximating net $CO_2$ fluxes in aquatic areas can we reproduce the observed $\Delta CO_2$ magnitude in both early and late cold season. The resulting seasonal change between the early and late cold season is consistent with the extended duration of the observed regional-scale zero curtain. The simplistic approximations suggested here are not inconsistent with the existing uncertainties in tundra $CO_2$ flux modeling and demonstrate the importance of considering these additional $CO_2$ fluxes and their mechanisms for future study.

## 4.3 Future state of net $CO_2$ flux on the Alaska North Slope

As the Arctic warms rapidly, the competition between the growing and cold season Arctic $CO_2$ fluxes will determine the net biogenic $CO_2$ flux into the atmosphere. Warming $T_a$ warms soils, thaws permafrost, increases active layer thickness and has extended the duration of the zero curtain from weeks to over 100 days (Romanovsky and Osterkamp, 2000; Schuur et al., 2015; Zona et al., 2016), all of which increase cold season $CO_2$ emissions. The warming may also increase net growing season uptake, but the severe light limitation at high northern latitudes limits the extent of the growing season, especially on the North Slope (Zhang et al., 2020). The future of $CO_2$ fluxes from inland waters and wetlands in the Arctic is uncertain, but some studies suggest $CO_2$ emissions from lakes may increase (Bayer et al., 2019). The culmination of these effects will likely push the North Slope into a consistent net source in the future. However, observations at the NOAA BRW tower during our study period do not show elevated late cold season $CO_2$ emissions, so the North Slope was not a consistent net source through 2017. Accordingly, care must be taken to accurately represent $CO_2$ fluxes from Arctic ecosystems during both the early and late cold season when calculating the annual net $CO_2$ budget. TVPRM could be used with projections of meteorology and SIF to calculate the future net $CO_2$ balance for this region, but we caution against overuse of the model using current parameters, as the flux-driver relationships in the rapidly warming Arctic ecosystems are changing so quickly that we would not assume accuracy into the future. While we can constrain the annual net $CO_2$ budget with existing data, the Arctic is rapidly changing and needs constant monitoring. The following recommendations would provide more detailed spatial and seasonal constraints and up-to-date information on the processes driving $CO_2$ fluxes across the region.

### 4.3.1 Future observation efforts

Our results motivate the need for a more extensive network of $CO_2$ eddy flux towers operating year-round, alongside sensors for soil moisture and $T_s$ profiles throughout the active layer to better understand the mechanisms driving year-round and especially early cold season $CO_2$ fluxes. Noting that automated or semi-automated monitoring systems for aquatic environments currently do not exist for the North Slope or other high latitude regions, this sensor network should be distributed throughout poorly sampled ecosystem types, particularly along wetness gradients that span mixed terrestrial-aquatic environments. The results in this study also support the need for additional continuous $CO_2$ concentration measurements at tall towers across the North Slope (including away from the coast) to increase coverage of observed $\Delta CO_2$ during all seasons and to better constrain the regional background. Airborne measurements of both $CO_2$ concentrations and $CO_2$ fluxes remain valuable to sample areas less accessible via ground-based measurements, but a large-scale flight campaign in the region has not occurred since 2017. Any additional flights should be targeted as early before, and as late after, the growing season as possible. Satellites that rely on reflected sunlight to detect $CO_2$ have increasingly been used to constrain $CO_2$ budgets in the northern latitudes (e.g., Byrne et al., (2022)), but data is very limited in the cold season, especially in far-northern regions like the North Slope.

### 4.3.2 Future modeling efforts

The large initial range of potential regional net $CO_2$ flux values we found for the Alaska North Slope indicates a large sensitivity to choices and assumptions made when scaling eddy flux observations from the site- to regional- scale. The most important of these choices are the representation of the upland tundra, particularly for the response of $R_{soil}$ to $T_s$ during the cold season, and the distribution of vegetation types throughout the domain. Future tundra $CO_2$ modeling efforts should focus on using site-level data that is the most consistent with regional-scale fluxes, rather than incorporating data from all available sites. Consistency and accuracy in classification schemes used in vegetation maps must also be addressed. As we have shown with the atmospheric observations, not all model scenarios have equal likelihood to be true, and the mean of the model ensemble is not necessarily the most likely or most consistent with the atmosphere. Using these atmospheric observations is uncertain, however, due to potential errors in the transport modeling, which are difficult to quantify. Atmospheric modeling of remote areas such as the Alaska North Slope requires further evaluation and improvement. Further, increasing model temporal resolution should be considered as the importance of the zero-curtain and snow cover to the net $CO_2$ flux of tundra ecosystems is recognized, both of which vary on the order of days and weeks, rather than months.

### 5. Conclusions

Observed atmospheric concentrations from aircraft and towers are a powerful tool that provide a regional constraint on the many combinations of possible $CO_2$ flux parameterizations and distributions of tundra ecosystems on the North Slope of Alaska. We find that the annual regional net $CO_2$ flux on the North Slope in not a consistent net source or sink, but instead

varies between –6 and +6 TgC yr$^{-1}$ for 2012–2017. We can also identify ecosystem relationships and driver combinations that
best represent both local $CO_2$ flux patterns and regional atmospheric $CO_2$ enhancements. The simulated regional net $CO_2$ flux
is highly sensitive to assumptions made while scaling up eddy flux observations, especially the ecosystem response to $T_s$ of
tundra during the cold season and the spatial distribution of tundra types across the North Slope. Additionally, scaling methods
that average observations from multiple eddy covariance flux sites should consider which sites are most representative of the
regional impact of the biosphere on the atmosphere using integrative top-down observations.

This work shows that year-round measurements of atmospheric $CO_2$ concentrations and fluxes across heterogeneous

terrestrial and aquatic ecosystems are needed to represent the drivers of $CO_2$ fluxes from Arctic regions. Arctic ecosystems
have the potential to accelerate warming if vast stores of carbon are released or buffer warming if increasing carbon uptake
from vegetation occurs. All components of Arctic tundra ecosystems must be fully incorporated into earth system models to
improve projections of future climate warming and associated carbon cycle feedbacks.
**Data availability**
Data that support the findings of this study are available as listed below:
TVPRM NEE for all ensemble simulations: https://doi.org/10.3334/ORNLDAAC/1920.
ICS, ICT, and ICH eddy flux tower observations: http://aon.iab.uaf.edu/data.
IVO, ATQ, BES, BEO, and CMDL eddy flux tower observations: https://doi.org/10.18739/A2X34MS1B.
NOAA BRW tower observations: https://www.esrl.noaa.gov/gmd/dv/data/?site=brw.
ARM-ACME V aircraft observations:  https://www.osti.gov/dataexplorer/biblio/dataset/1346549.
ABoVE Arctic-CAP aircraft observations:  https://doi.org/10.3334/ORNLDAAC/1658.
NARR meteorology: https://psl.noaa.gov/data/gridded/data.narr.html.
ERA5 meteorology: https://www.ecmwf.int/en/forecasts/dataset/ecmwf-reanalysis-v5.
GOME-2 SIF: https://avdc.gsfc.nasa.gov/pub/data/satellite/MetOp/GOME_F/.
GOSIF: https://globalecology.unh.edu/data/GOSIF.html.
CSIF: http://doi.org/10.6084/m9.figshare.6387494.
CAVM vegetation map: https://www.geobotany.uaf.edu/cavm/.
RasterCAVM vegetation map: https://dx.doi.org/10.17632/c4xj5rv6kv.1.
ABoVE LC vegetation map: https://doi.org/10.3334/ORNLDAAC/1691.
RS-PM $T_s$: available from authors upon request.
NOAA BRW tower and ARM-ACME V aircraft campaign WRF-STILT footprints:
https://doi.org/10.3334/ORNLDAAC/1431, particle trajectories: https://doi.org/10.3334/ORNLDAAC/1430.
ABoVE Arctic-CAP aircraft campaign WRF-STILT footprints: https://doi.org/10.3334/ORNLDAAC/1896, particle
trajectories: https://doi.org/10.3334/ORNLDAAC/1895.
Luus et al. (2017) fluxes: https://doi.org/10.3334/ORNLDAAC/1314.
Commane et al. (2017) optimized fluxes: https://doi.org/10.3334/ORNLDAAC/1389.
Natali and Watts et al. (2019) fluxes: https://doi.org/10.3334/ORNLDAAC/1683.
Watts et al. (2021) fluxes: https://doi.org/10.3334/ORNLDAAC/1935.

**Author contributions**

LDS and RC designed the study. KAA, ESE, JPG, AK, WCO, and DZ provided eddy covariance flux tower data. SCB, KM,
and CS provided aircraft concentration data. JMH and MEM provided WRF-STILT particle files and footprints. YY provided
RS-PM $T_s$ data. JDW provided Watts et al. (2021) cold season belowground $CO_2$ fluxes. LDS developed and evaluated
TVPRM net $CO_2$ fluxes against observations. RC, EJLL, JWM, and JDW assisted the analysis. LDS wrote the paper. All co-
authors contributed to the preparation of the manuscript.

**Competing interests**

Authors declare that they have no competing interests.

**Acknowledgments**

We would like to acknowledge that the Alaskan North Slope is home to multiple Alaska Native nations, including the
Nunamiut, Gwich'in, Koyukuk, and Iñupiaq peoples. We support and honor the place-based knowledge of Indigenous Peoples
and recognize their ancestral and contemporary stewardship of their homelands that we research. LDS and RC are supported
by research funding from the Department of Earth and Environmental Sciences at Columbia University and the NASA ABoVE
grant #NNX17AC61A. LDS is addtionally supported by the National Science Foundation (NSF) Office of Polar Programs
grant #1848620. EJRL and JWM are supported by NASA ABoVE grant #NNX17AE75G. JDW is supported by NASA
ABoVE grant #80NSSC19M0209 and NASA grant #NNH17ZDA001N-NIP. Part of the research was carried out at the Jet
Propulsion Laboratory, California Institute of Technology, under a contract with NASA (80NM0018D0004). Imnavait Creek
flux towers are funded under grants from the NSF Office of Polar Programs, 1503912 and 0632264. Resources supporting
JMH and WRF-STILT modeling were provided by NASA grant #NNX17AE75G, #NNX17AC61A, and the NASA High-End
Computing (HEC) Program through the NASA Advanced Supercomputing (NAS) Division at Ames Research Center. We
thank the R Project community for analysis and plotting tools, especially the ggplot2, ggpattern, magick, anytime, lubridate,
raster, and cowplot packages. NCEP Reanalysis data provided by the NOAA/OAR/ESRL PSL, Boulder, Colorado, USA.
Some of the data products used in this paper were acquired for the CARVE, a NASA Earth Ventures Sub-orbital (EV-S1)
investigation.

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
