# Peer review of "Using atmospheric observations to quantify annual biogenic carbon dioxide fluxes on the Alaska North Slope"

_Biogeosciences, 2022_

## Referee Comment (RC1)

**Review of "Using atmospheric observations to quantify annual biogenic carbon dioxide fluxes on the Alaska North Slope" by *Schiferl et al.* (2022)**

The manuscript by *Schiferl et al.* (2022) integrates atmospheric and ground in situ observations, remote-sensing data, and the Tundra Vegetation Photosynthesis and Respiration Model (TVPRM) ensemble, which was developed in this study, to quantify the annual net biospheric carbon dioxide ($CO_2$) flux and seasonality from the North Slope of Alaska. Using observations to optimize TVPRM predictions, it was determined that the North Slope is a near-neutral flux of $CO_2$ (ranging between -6 to +6 TgC $yr^{-1}$). The interannual variability of the net $CO_2$ flux from this region varied between a small source and sink of carbon to the atmosphere and is driven be yearly differences in the strength of the $CO_2$ uptake growing season. The non-growing season is shown to be a large source of $CO_2$ to the atmosphere driven by soil respiration and inland aquatic systems during the early cold season which counteracts the carbon sink during the summer months. However, this work did not find the large late cold season $CO_2$ respiration in this region that has been identified in other recent studies. This work demonstrates that there are numerous uncertainties in the capability to upscale observations to regional-scale net $CO_2$ flux estimates and suggests that higher spatiotemporal observation coverage is needed to improve the accuracy of net $CO_2$ flux estimates from the North Slope in the present and the future.

The study by *Schiferl et al.* (2022) applies an impressive amount of data sets to derive net $CO_2$ flux estimates for the North Slope between 2012 and 2017. The TVPRM predictions are optimized using atmospheric $CO_2$ measurements and an atmospheric transport model and the TVPRM predictions are compared to other estimates for this region. These aspects and the comprehensive evaluation of TVPRM predictions are impressive aspects of the study. However, the text itself is challenging to follow in multiple parts of the manuscript and could be improved with some rewriting. There are a large number of figures (which themselves have numerous sub-panels/legends and dense figure captions) and tables in the main body of the manuscript and the supplemental information section which the authors bounce back and forth between throughout the paper. The manuscript presentation and readability could be improved by some reorganization and simplification. Furthermore, I feel that the paper lacks discussion about the novel aspects of the work and how it advances the scientific understanding of the field. These issues, along with some potential issues with the methods and interpretations of the results of this study, are described further below. With some major revisions and improvements in the writing of the text, I think this paper could be published in *Biogeosciences*.

**Major Comments**

1. More attention and details in the text are needed when describing how TPVRM variable parameters are derived. Text S1 should be expanded and potentially placed in the main text of the manuscript. First off, statistics on the correlation between observed values of $CO_2$ flux and $T_s/T_a$ to the $\alpha_s/\alpha_a$ and $\beta_s/\beta_a$ fitting parameters should be presented in the text of Step 1 and Step 2, respectively. Same thing for the non-linear fits derived in Step 3. Secondly, median observed net

$CO_2$ fluxes are used for the linear fits in Step 1 and 2; however, the instantaneous 30-min observed net $CO_2$ flux data are used in Step 3. Why are the observed $CO_2$ values treated differently in these steps? Also, are the median values for Step 1 and 2 determined for the entire 365 day moving window? Finally, many constants are presented (e.g., $PAR_0$, initial $\lambda$, % of potential growing and non-growing days needed, % of half-hourly $CO_2$ observations that are negative, etc.) throughout Text S1 that have no references or justification/explanation of why they were chosen. These mentioned aspects, and any others the authors think could improve the description of how TVPRM fits are derived, need to be expanded upon in the revised manuscript.

2. How are $CO_2$ fluxes from sources other than the terrestrial biosphere accounted for in observations of $CO_2$ enhancements ($\Delta CO_2$)? The tall-tower and aircraft measurements observe total $CO_2$ from all flux sources including regional fossil fuel usage, waste burning, shipping, or small fires not removed "by elevated or varying carbon monoxide (CO) concentrations". Exactly how CO was used for the purpose of removing the influence of wildfires needs to be better explained. Overall, if $\Delta CO_2$ from all the other sources of $CO_2$ in this region are not removed from the observations, the comparison between them and simulated values will be biased for incorrect reasons. This needs to be better described in the text.

3. The organization of the paper made it a challenge to read. For instance, Fig. 2 "Constrained" TVPRM predictions are shown here in the results. It was not easy to follow what the constrained TVPRM values were. Reading further, much past where Fig. 2 is discussed, I see on Line 345 this explanation is provided. It would be best if the discussion of the model performance and clearer description of how the "best" model ensemble members were determined is needed in the methods section (before results are being discussed). Furthermore, ZC and IW are finally described in Sect. 3.4 after being introduced well after they are being shown in the results. This made interpreting a large portion of the paper very difficult.

This brings up a larger point. The paper itself is very dense when including the supplementary material which includes 18 additional figures all of which include numerous sub-panels. The text jumps between supplementary figures and the main text very frequently which makes interpreting the work difficult. Is there a way to reorganize the text and potentially reduce the number of figures (all of which have many different panels, titles, legends, and very dense captions) and tables to streamline the study?

4. Accuracy of Weather Research and Forecasting (WRF) meteorology over the North Slope and BRW tower. This is an aspect which is not discussed in the study and could potentially be very important for the results and interpretation of this work. How well does WRF capture the winds (speed and direction) over the region and at BRW? How about planetary boundary layer (PBL) dynamics in this region? Are there meteorological stations, or aircraft observations, which could be used to assess the WRF winds and PBL prediction accuracy? Biased WRF simulations will bias the comparison of observed and simulated $\Delta CO_2$ values. This could be one of the main reasons why TVPRM in this study, and other past net $CO_2$ flux estimate products do not capture the magnitudes and seasonality of $\Delta CO_2$ at BRW. This tower is located on the coast, and it is possible

that the model is not performing well in this location. I don't think this paper can be published without providing some demonstration about the accuracy of the WRF meteorology used in this study.

5. WRF model set up. There is no mention about details of the WRF model setup used to derive the atmospheric transport and surface sensitivity footprints applied in this study. What is the horizontal and vertical resolution of the WRF model used? What version is applied? How many spatial domains were used in the simulations? What physics options (e.g., schemes for long- and short-wave radiation, microphysics, convection, PBL, land surface, etc.) were selected for the model simulations? The differences in WRF setups can directly impact the accuracy of the model predictions.

6. Line 399-410. Beyond the fact that it improves the comparison of simulated $\Delta CO_2$ values to observations at BRW, why is the constant 0.25 $\mu mol\ m^{-2}\ s^{-1}$ zero-curtain emission source applied for October, which decreases to zero in December, chosen to add to TVPRM constrained estimates? Are there any past studies which could justify adding this value? Some justification needs to be provided for why these zero-curtain emission values were chosen.

Also, more detail is needed to why the coastal tundra ecosystem parameterization was applied for inland aquatic fluxes. What inland water map was used to derive the location of all inland water bodies? Is lake ice phenology considered when estimating inland aquatic fluxes? How much $CO_2$ is estimated to be emitted, or absorbed, by lakes throughout the year using these methods? In reality, lakes will have very little open water interaction with the atmosphere in the cold season as they can be frozen in this region.

7. Line 444-446. Net Annual $CO_2$ flux. The largest annual uptake of $CO_2$ between 2012 and 2017 was during 2013 and 2015. What was different about these years compared to the others in this time period? Is there a strong correlation with soil/air temperature, precipitation, snowpack, etc.? How about wildfires? From first glance it appears that these two years had the most acreage burned by fires in Alaska during the time period studied here ([https://uaf-iarc.org/alaskas-changing-wildfire-environment/](https://uaf-iarc.org/alaskas-changing-wildfire-environment/)). The text describes that the balance of $R_{soil}$, $R_{plant}$, and GPP control the overall biospheric $CO_2$ flux; however, some description of the controlling variables on interannual variability of net $CO_2$ flux in this region would improve the scientific impact of this study.

8. What are the scientific advancements of this study? The work does a nice job of combining in situ and remote-sensing data and models to estimate the annual net $CO_2$ flux from the North Slope of Alaska. However, beyond the detailed description of how the TVPRM estimates were optimized to match atmospheric observations, what is the importance of the TVPRM model development? A near neutral net annual $CO_2$ flux for the North Slope is derived with TVPRM which is said to be consistent with past model ensemble estimates (Fisher et al., 2014), so this result is really only novel compared to some past estimates from Luus et al. (2017), Natali et al. (2019), and Watts et al. (2021) discussed in the text. An interesting finding is the TVPRM prediction of interannual variability of $CO_2$ fluxes in the region. The fact that the model suggested the net annual $CO_2$ flux changes between small sources and sinks is interesting. The study states that variability in uptake

season strength drives this variability; however, what are the physiochemical variables driving these differences? Is it precipitation, snowpack, air/soil temperature, fires, etc.? There is a lot that could be studied here to improve the novel aspects of the work. Looking into these physiochemical drivers, and their control on net $CO_2$ fluxes, would really help the reader understand what controlling variables could drive future changes in this region. This was stated in the text to be an importance of this work but really isn't addressed here at all.

9. Could the TVPRM model be used with future gridded predictions of meteorology, vegetation, hydrology, and other sources of information to predict future changes in the net $CO_2$ flux of the North Slope? If so, it is likely beyond this study to do so, but this should be discussed in the conclusions section of the text to increase the scientific impact of this work.

10. Vegetation maps. A major finding in this work is that vegetation distributions and ecosystem type information is a controlling factor on the ability to accurately model $CO_2$ fluxes in this region. Are the three vegetation maps used in this study (CAVM, RasterCAVM, ABoVE LC) the only ones available for this region? If there are other vegetation maps available, why aren't they used in this study since it is very important for TVPRM $CO_2$ flux calculation accuracy? If there are no other maps of vegetation distributions and ecosystem type, how should CAVM, RasterCAVM, and ABoVE LC be improved to assist improvement in $CO_2$ flux calculation accuracy?

11. Could the results from TVPRM be compared to net $CO_2$ flux estimates from other terrestrial biosphere models (e.g., CASA, SiB4, Jules, Orchidee, etc.) in this region? Does TVPRM improve upon these established terrestrial biosphere models?

**Minor Comments**

1. Line 28. Not sure what "top-down" observations at atmospheric $CO_2$ are. Do the authors mean top-down emission estimates using atmospheric observations? In Sect. 2.3 I think top-down observations of $CO_2$ *enhancements* ($\Delta CO_2$) is the correct way to use this term. This occurs at other locations throughout the manuscript. Observations of concentrations are themselves not typically classified as top-down, but enhancements and emission estimates using models and the observations are more often termed as top-down.

2. Line 78-9. IVO, CMDL, TVPRM, CSIF, and SIF have yet to be defined in the text.

3. Line 174-175. More appropriate to reference Lin et al. (2003) for WRF-STILT.

4. Figure S1. Are the multi-colored lines in each panel of Fig. S1 the "Lines for matching site parameters and locations are highlighted"? This needs to be described more clearly either in the figure caption or in the text. I had a very difficult time understanding what these lines represented.

5. Figure S4. This figure has a lot of information in it yet is only introduced in the text. Can the authors describe the performance of the model in more detail? A couple sentences discussing inter-site performance and the differences between seasons and averaging time periods would be helpful as there are very large differences which would be of interest to the reader.

6. Line 318-321. In Fig. 3b the reader can not distinguish between the early and late cold season as discussed in the text. Only in Fig. S11 is the temporal color scaled used.

7. Fig. S14. To compare TVPRM Constrained with RS-PM $T_{soil}$ to the TVPRM Constrained using NARR data the text needs to reference which figure and sub-panel to compare Fig. S14a to.

8. Line 398. Missing "the" in this sentence.

9. Line 56 and throughout. Why is Natali et al. (2019) referenced as Natali & Watts et al., 2019? Also, "&" and "and" are interchangeable used throughout the paper. Might be better to choose one for consistency.

10. Line 516. "motivate" instead of "motive".

**References**

Fisher, J. B., Sikka, M., Oechel, W. C., Huntzinger, D. N., Melton, J. R., Koven, C. D., Ahlström, A., Arain, M. A., Baker, I., Chen, J. M., Ciais, P., Davidson, C., Dietze, M., El-Masri, B., Hayes, D., Huntingford, C., Jain, A. K., Levy, P. E., Lomas, M. R., Poulter, B., Price, D., Sahoo, A. K., Schaefer, K., Tian, H., Tomelleri, E., Verbeeck, H., Viovy, N., Wania, R., Zeng, N., and Miller, C. E.: Carbon cycle uncertainty in the Alaskan Arctic, Biogeosciences, 11, 4271‑4288, https://doi.org/10.5194/bg-11-4271-2014, 2014.

Lin, J. C., Gerbig, C., Wofsy, S. C., Andrews, A. E., Daube, B. C., Davis, K. J., and Grainger, A.: A near-field tool for simulating the upstream influence of atmospheric observations: the Stochastic Time-Inverted Lagrangian Transport Model (STILT), J. Geophys. Res., 108, 4493, doi:10.1029/2002JD003161, 2003.

Luus, K. A., Commane, R., Parazoo, N. C., Benmergui, J., Euskirchen, E. S., Frankenberg, C., Joiner, J., Lindaas, J., Miller, C. E., Oechel, W. C., Zona, D., Wofsy, S., and Lin, J. C.: Tundra photosynthesis captured by satellite-observed solar-induced chlorophyll fluorescence, Geophys. Res. Lett., 44, 2016GL070842, https://doi.org/10.1002/2016GL070842, 2017.

Natali, S. M., Watts, J. D., Rogers, B. M., Potter, S., Ludwig, S. M., Selbmann, et al.: Large loss of $CO_2$ in winter observed across the northern permafrost region, Nat. Clim. Change, 9, 852‑857, https://doi.org/10.1038/s41558-019-0592-8, 2019.

Watts, J. D., Natali, S. M., Minions, C., Risk, D., Arndt, K., Zona, D., Euskirchen, E. S., Rocha, A. V., Sonnentag, O., Helbig, M., Kalhori, A., Oechel, W., Ikawa, H., Ueyama, M., Suzuki, R., Kobayashi, H., Celis, G., Schuur, E. A. G., Humphreys, E., Kim, Y., Lee, B.-Y., Goetz, S., Madani, N., Schiferl, L. D., Commane, R., Kimball, J. S., Liu, Z., Torn, M. S., Potter, S., Wang, J. A., Jorgenson, M. T., Xiao, J., Li, X., and Edgar, C.: Soil respiration strongly offsets carbon uptake in Alaska and Northwest Canada, Environ. Res. Lett., 16, 084051, https://doi.org/10.1088/1748-9326/ac1222, 2021.

---

## Author Comment (AC1)

**RC1**
**Review of "Using atmospheric observations to quantify annual biogenic carbon dioxide fluxes on the Alaska North Slope" by Schiferl et al. (2022)**

The manuscript by Schiferl et al. (2022) integrates atmospheric and ground in situ observations, remote-sensing data, and the Tundra Vegetation Photosynthesis and Respiration Model (TVPRM) ensemble, which was developed in this study, to quantify the annual net biospheric carbon dioxide ($CO_2$) flux and seasonality from the North Slope of Alaska. Using observations to optimize TVPRM predictions, it was determined that the North Slope is a near-neutral flux of $CO_2$ (ranging between -6 to +6 TgC yr$^{-1}$). The interannual variability of the net $CO_2$ flux from this region varied between a small source and sink of carbon to the atmosphere and is driven be yearly differences in the strength of the $CO_2$ uptake growing season. The non-growing season is shown to be a large source of $CO_2$ to the atmosphere driven by soil respiration and inland aquatic systems during the early cold season which counteracts the carbon sink during the summer months. However, this work did not find the large late cold season $CO_2$ respiration in this region that has been identified in other recent studies. This work demonstrates that there are numerous uncertainties in the capability to upscale observations to regional-scale net $CO_2$ flux estimates and suggests that higher spatiotemporal observation coverage is needed to improve the accuracy of net $CO_2$ flux estimates from the North Slope in the present and the future.

The study by Schiferl et al. (2022) applies an impressive amount of data sets to derive net $CO_2$ flux estimates for the North Slope between 2012 and 2017. The TVPRM predictions are optimized using atmospheric $CO_2$ measurements and an atmospheric transport model and the TVPRM predictions are compared to other estimates for this region. These aspects and the comprehensive evaluation of TVPRM predictions are impressive aspects of the study. However, the text itself is challenging to follow in multiple parts of the manuscript and could be improved with some rewriting. There are a large number of figures (which themselves have numerous subpanels/legends and dense figure captions) and tables in the main body of the manuscript and the supplemental information section which the authors bounce back and forth between throughout the paper. The manuscript presentation and readability could be improved by some reorganization and simplification. Furthermore, I feel that the paper lacks discussion about the novel aspects of the work and how it advances the scientific understanding of the field. These issues, along with some potential issues with the methods and interpretations of the results of this study, are described further below. With some major revisions and improvements in the writing of the text, I think this paper could be published in Biogeosciences.

We thank the reviewer for their thorough and helpful comments and suggestions. We have done our best to balance the requests for additional information and details while also keeping the paper from becoming additionally complicated. Specific responses to each comment follow in red, with proposed edits to the manuscript in blue. Line numbers refer to the original manuscript.

**Major Comments**

1. More attention and details in the text are needed when describing how TPVRM variable parameters are derived. Text S1 should be expanded and potentially placed in the main text of the manuscript. First off, statistics on the correlation between observed values of $CO_2$ flux and $T_s/T_a$ to the $\alpha_s/\alpha_a$ and $\beta_s/\beta_a$ fitting parameters should be presented in the text of Step 1 and Step 2, respectively. Same thing for the non-linear fits derived in Step 3. Secondly, median observed net $CO_2$ fluxes are used for the linear fits in Step 1 and 2; however, the instantaneous 30-min observed net $CO_2$ flux data are used in Step 3. Why are the observed $CO_2$ values treated differently in these steps? Also, are the median values for Step 1 and 2 determined for the entire 365 day moving window? Finally, many constants are presented (e.g., $PAR_0$, initial $\lambda$, % of potential growing and non-growing days needed, % of half-hourly $CO_2$ observations that are negative, etc.) throughout Text S1 that have no references or justification/explanation of why they were chosen. These mentioned aspects, and any others the authors think could improve the description of how TVPRM fits are derived, need to be expanded upon in the revised manuscript.

As the reviewer points out the existing complexity of the paper, we choose to keep the details of the procedure for deriving the TVPRM variable parameters in Sect. S1 of the Supplement. The main components of the procedure (linear regression to determine respiration components, non-linear regression to determine GPP components) largely follow that of the previous iteration of this empirical $CO_2$ flux model, PVPRM-SIF, in Luus et al., (2017). However, instead of using snow cover as the indicator of $T_a$-driven respiration (no snow) or $T_s$-driven respiration (snow), we separate respiration into $R_{soil}$ and $R_{plant}$ components, which explicitly represent heterotrophic and autotrophic respiration communities, respectively. $R_{soil}$ is now applied year-round, with $R_{plant}$ applied during the growing season as determined by SIF. This change also simplifies the required model inputs to only reanalysis data and SIF.

Given that we determine the parameters for each site using a moving-window approach, it is impractical to present all the statistics for each of these fits (N = 7132). Instead, we present the statistics on the site-level observation/simulation comparison for net $CO_2$ flux using the median parameters (used later to scale to the regional domain) in Fig. S4, which are more representative of the ultimate performance of the model. The results of this comparison are described in Sect. S4. We now point to this comparison toward the end of Sect. S1.

For step 1, we use daily mean $T_s$ and daily mean observed net $CO_2$ flux (Supplement line 16) to account for the lack of variability in input $T_s$ from reanalysis products on sub-daily timescales. For steps 2 and 3, we use half-hourly $T_a$ and PAR and the corresponding half-hourly observed net $CO_2$ flux (Supplement lines 20, 24-25, respectively) as these variables have considerable diurnal variability. The "median observed net $CO_2$ flux" mentioned in steps 1 and 2 refers to the results of the 5% binning employed prior to the regression calculation for that 365-day window. This binning by ordered $T_s$ and $T_a$ (and their corresponding observed net $CO_2$ flux) in each step more evenly distributes the influence of high- and low-end values in the regression. For $T_s$, the distribution of values is non-normal, with a majority of points just below 0°C during the long zero-curtain period. For $T_a$, the distribution of values is sporadic and variable as data from the light-limited growing season is limited to August and the number of total points available is only ~10% of those used in the $R_{soil}$ fit.

We have clarified steps 1 and 2 in Sect. S1 to read as follows:

"Step 1: *Linear regression of observed net $CO_2$ flux against soil temperature ($T_s$) during non-growing season to determine $\alpha_s$ and $\beta_s$ and calculate soil respiration ($R_{soil}$).* Daily mean $T_s$ and the corresponding daily mean observed net $CO_2$ flux during potential non-growing days (daily maximum air temperature ($T_a$) < 0°C) when SIF = 0 and 50% of the half-hours have observed net $CO_2$ flux are identified and sorted into 5% bins by ordering the daily mean $T_s$. Regression is performed on the 20 median observed net $CO_2$ flux and $T_s$ values calculated from these bins. Daily values are used here to account for the lack of variability in $T_s$ from reanalysis products on sub-daily timescales. The binning approach distributes the influence of low-end $T_s$ values more evenly in the regression, which is needed because the distribution of $T_s$ values is non-normal, with a majority of points just below 0°C during the long zero-curtain period."

"Step 2: *Linear regression of observed net $CO_2$ flux against $T_a$ during growing-season night to determine $\alpha_a$ and $\beta_a$ and calculate plant respiration ($R_{plant}$).* Half-hourly $T_a$ and the corresponding half-hourly observed net $CO_2$ flux with $R_{soil}$ (calculated in step 1) removed during potential growing days (daily minimum $T_a$ > 0°C) when solar-induced chlorophyll fluorescence (SIF) > 0 and photosynthetically active radiation (PAR) <= 4 μmol photon m$^{-2}$ s$^{-1}$ are identified and sorted into 5% bins by ordering the half-hourly $T_a$. Regression is performed on the 20 median observed net $CO_2$ flux with $R_{soil}$ removed and $T_a$ values calculated from these bins. The binning approach distributes the influence of $T_a$ values more evenly in the regression, which is needed because distribution of values is sporadic and variable as data from the light-limited growing season is limited to August and the number of total points available is only ~10% of those used in the $R_{soil}$ fit."

The initial values used for the nls in step 3 (PAR0 = 240 and λ = 0.04) come from the shrub tundra parameters reported by Luus et al. (2017). Other criteria such as % of potential growing and non-growing days needed, % of half-hourly CO2 observations that are negative were chosen to balance maintaining representativeness of the fit (i.e., having data from throughout the entire time period) and keeping enough data to be useful for a stable fit (i.e., non-growing season data is more limited). The criteria for the TPVRM model data filtering and tuning described here also results in the best version of the model compared to observations after many iterations and rounds of testing.

We have added the following to the end of Sect. S1 to clarify several of the above points:

"The main components of the procedures for steps 1-3 above (i.e., linear regressions for respiration, non-linear regression for GPP) largely follow that of the previous version of this empirical $CO_2$ flux model described by Luus et al., (2017). However, instead of using snow cover as the indicator of $T_a$-driven total respiration (no snow) or $T_s$-driven total respiration (snow), as in Luus et al., (2017), we separate respiration into $R_{soil}$ and $R_{plant}$ components, which explicitly represent heterotrophic and autotrophic respiration communities, respectively. $R_{soil}$ is now applied year-round, with $R_{plant}$ applied during the growing season as determined by SIF. This change also simplifies the required model inputs to only reanalysis data and SIF.

The threshold criteria described above for performing a regression calculation during a particular window and for filtering data used in the regressions were chosen to balance maintaining representativeness of the various regressions (i.e., data is available from throughout the entire time period) and keeping enough data to be useful for a stable fit (i.e., non-growing season data is more limited). The methods for determining the TPVRM parameters described

here also result in the best version of the model compared to observations after many development iterations."

2. How are $CO_2$ fluxes from sources other than the terrestrial biosphere accounted for in observations of $CO_2$ enhancements ($\Delta CO_2$)? The tall-tower and aircraft measurements observe total $CO_2$ from all flux sources including regional fossil fuel usage, waste burning, shipping, or small fires not removed "by elevated or varying carbon monoxide (CO) concentrations". Exactly how CO was used for the purpose of removing the influence of wildfires needs to be better explained. Overall, if $\Delta CO_2$ from all the other sources of $CO_2$ in this region are not removed from the observations, the comparison between them and simulated values will be biased for incorrect reasons. This needs to be better described in the text.

In addition to the biosphere, other potential sources of $CO_2$ on the Alaska North Slope include biomass burning and anthropogenic activity. Together, these other sources are small and regionally contribute less than 1 TgC to the atmosphere for our study period, according to EDGAR anthropogenic and GFED biomass burning inventories (see below). Biomass burning is highly variable from year to year. Even during high fire years for the entirety of Alaska, such as 2015, there has been little fire activity on the North Slope.

The following was added to the Supplement as Table S1:

Table S1. Annual and seasonal $CO_2$ emission totals from anthropogenic and biomass burning sources and area burned in the Alaska North Slope and all of Alaska for 2012–2017. Annual anthropogenic emissions are from EDGAR, the Emissions Database for Global Atmospheric Research v7.0 (https://edgar.jrc.ec.europa.eu/dataset_ghg70). Monthly biomass burning emissions are from GFED, Global Fire Emissions Database v4 (https://globalfiredata.org/pages/data/#emissions). Area burned data is from the Alaska Interagency Coordination Center via UAF SNAP tool (https://snap.uaf.edu/tools/daily-fire-tally).

| | Domain | 2012 | 2013 | 2014 | 2015 | 2016 | 2017 | Jun-Sep 2015 | May-Nov 2017 |
|---|---|---|---|---|---|---|---|---|---|
| Anthropogenic $CO_2$ Emissions [TgC] | North Slope | 0.73 | 0.74 | 0.78 | 0.82 | 0.77 | 0.79 | | |
| | Alaska | 7.7 | 7.7 | 7.8 | 8.2 | 8.3 | 8.4 | | |
| Biomass Burning $CO_2$ Emissions [TgC] | North Slope | 0.23 | 0.12 | 0.00 | 0.12 | 0.34 | 0.07 | 0.12 | 0.07 |
| | Alaska | 0.97 | 6.7 | 1.7 | 28 | 1.9 | 7.6 | 28 | 7.6 |
| Area Burned [million acres] | Alaska | | 1.3 | | 5.1 | 0.50 | 0.65 | 5.1 | 0.65 |

We note that area burned values for 2012 and 2014 data were not available but are lower than 2016 in the figure reference the reviewer at https://uaf-iarc.org/alaskas-changing-wildfire-environment/.

Domains used in above quantification:

[Figure]

As the reviewer notes, to avoid comparing our model to atmospheric observations of non-biogenic sources, we remove observational time periods from our analysis whenever "indicated by elevated or varying carbon monoxide (CO) concentrations" (line 122). More specifically, $CO_2$ data with corresponding CO concentrations greater than 150 ppb are removed as in Chang et al. (2014) and Commane et al. (2017), which indicates a strong local combustion source. Observational time periods with variable CO concentrations, as indicated by 40 ppb change throughout a profile or horizontal transit, which may not meet the 150 ppb threshold, indicate complex mixing of more remote combustion sources and are also removed from the analysis (Chang et al., 2014).

We have clarified the description of the CO-filtering and expectations of observing combustion $CO_2$ fluxes on the North Slope in Sect. 2.1.1 to read as follows:

"For the ARM-ACME V and ABoVE Arctic-CAP aircraft campaign observations, we group averaged sampling points into 50 m vertical bins after removing data influenced by combustion sources such as anthropogenic activity and biomass burning events. These combustion sources of $CO_2$ are expected to be small (<1 TgC yr$^{-1}$ on the North Slope, see Table S1) during our study period. They are not accounted for in biogenic $CO_2$ flux models, however, and must be removed from our analysis when observed. We remove time periods with elevated carbon monoxide (CO) concentration above 150 ppb, as in Chang et al. (2014) and Commane et al. (2017), which indicates local combustion sources. Time periods with highly variable CO concentrations ($\Delta CO > 40$ ppb) indicate complex mixing of more remote combustion sources and are also removed (Chang et al., 2014)."

3. The organization of the paper made it a challenge to read. For instance, Fig. 2 "Constrained" TVPRM predictions are shown here in the results. It was not easy to follow what the constrained TVPRM values were. Reading further, much past where Fig. 2 is discussed, I see on Line 345 this explanation is provided. It would be best if the discussion of the model performance and clearer description of how the "best" model ensemble members were determined is needed in the methods section (before results are being discussed). Furthermore, ZC and IW are finally described in Sect 3.4 after being introduced well after they are being shown in the results. This made interpreting a large portion of the paper very difficult.

The use of the atmospheric observations to constrain the TVPRM ensemble is a key result of the paper and would not be appropriate for the methods.

We have made the following changes to improve clarity of the paper organization and Fig. 2:

Added clarifications throughout the final paragraph of Sect. 2.5 to outline the organization of the results to better streamline the flow for the reader, now mentioning upcoming terminology (TVPRM Unconstrained, Constrained, ZC and IW) and the corresponding sections which describe them.

This paragraph now reads:
"These comparisons enable us to constrain the regional net CO2 flux on the Alaska North Slope. First, we identify the year-round empirically driven net CO2 fluxes from the TVPRM ensemble (TVPRM Unconstrained) which are most consistent with the CO2 concentration observations from the two aircraft campaigns and at the tower (TVPRM Constrained) (Sects. 3.1–3.2). Then, noting the large range in potential cold season CO2 fluxes, we compare our constrained TVPRM member with CO2 fluxes from previous studies (Sect. 3.3). Finally, we suggest and quantify sources of the missing CO2 flux observed during the early cold season (defined here as September–December) and incorporate those fluxes into our net CO2 budget (TVPRM Constrained + Additional Zero Curtain Emissions (ZC) and Inland Water Fluxes (IW)) (Sect. 3.4). This analysis provides a unique regional net CO2 flux quantification for the North Slope that is verified using atmospheric observations and can also be explained from an ecological and physical perspective."

It will only further complicate the paper to separate the figures into unconstrained and constrained versions, so we have added pointers in the Fig. 2 caption to clarify that the black points represent values from the constrained TVPRM with ZC and IW and that the description of this scenario comes later in the paper.

The caption now reads:
"Aircraft and tower $CO_2$ concentration measurements constrain year-round simulated $CO_2$ fluxes on the Alaska North Slope. **(a)–(c)** Comparison of observed and simulated $\Delta CO_2$ during the ARM-ACME V flight campaign **(a)**, during the ABoVE Arctic-CAP flight campaign **(b)**, and at the NOAA BRW tower **(c)** for air over the Alaska North Slope. Horizontal lines indicate range of uncertainty in the NOAA BRW tower ocean sector background calculation. Vertical boxes colored by month of the year represent 50% and whiskers represent 95% of $\Delta CO_2$ values from all members of unconstrained TVPRM ensemble (see Sect. 2.4) from all binned points. Black points show values from the constrained TVPRM member with additional zero-curtain emissions (ZC) and inland water fluxes (IW) (see Sect. 3.4). For **(a)–(b)**, observed values are vertically binned medians, and for constrained TVPRM member + ZC and IW, vertical lines contain middle 95% of $\Delta CO_2$ values from all binned points. **(d)** Combined comparison of observed and simulated $\Delta CO_2$ for all aircraft and tower points using constrained TVPRM member + ZC and IW. Shown with linear best fit (red line), slope determined by ordinary least squares, and coefficient of determination ($R^2$) of all points (n = 455). 1:1 line shown in dark gray."

This brings up a larger point. The paper itself is very dense when including the supplementary material which includes 18 additional figures all of which include numerous sub-panels. The text jumps between supplementary figures and the main text very frequently which makes interpreting the work difficult. Is there a way to reorganize the text and potentially reduce the number of figures (all of which have many different panels, titles, legends, and very dense captions) and tables to streamline the study?

Based on the above suggestions, we have strived to make the main text and science discussion as streamlined as possible given the numerous datasets and methods included in this study. The main text is structured straightforwardly with only five figures that focus on the atmospheric observations and quantifying the net $CO_2$ flux. The extensive supplementary material included is necessary to provide additional information to those interested in the topic who may find worth in the extra time it takes to distill those details, therefore we have not reduced the content of the Supplement.

4. Accuracy of Weather Research and Forecasting (WRF) meteorology over the North Slope and BRW tower. This is an aspect which is not discussed in the study and could potentially be very important for the results and interpretation of this work. How well does WRF capture the winds (speed and direction) over the region and at BRW? How about planetary boundary layer (PBL) dynamics in this region? Are there meteorological stations, or aircraft observations, which could be used to assess the WRF winds and PBL prediction accuracy? Biased WRF simulations will bias the comparison of observed and simulated $\Delta CO_2$ values. This could be one of the main reasons why TVPRM in this study, and other past net $CO_2$ flux estimate products do not capture the magnitudes and seasonality of $\Delta CO_2$ at BRW. This tower is located on the coast, and it is possible that the model is not performing well in this location. I don't think this paper can be published without providing some demonstration about the accuracy of the WRF meteorology used in this study.

Use of numerical weather prediction (NWP) models at high resolution (3.3km) permits more realistic depiction of the wind field compared to coarser NWP simulations (~10-30km) and both regional and global reanalysis products (~30-75km). This more realistic simulation is in part because of improved spatial representation of the underlying topography and land use. The model representation of hills and valleys, and coastlines, is also sharper than in other sources of meteorological fields. The magnitudes of extreme wind events associated with transient extratropical cyclones in the region, as well as flow through and over mountain ranges, is also much improved. Downslope windstorms that are absent in coarser-scale model grids are now resolved. The reviewer is referred to Henderson et al. (2015), which documented the WRF-STILT model configuration for high-latitude transport modeling during the NASA Carbon in Arctic Reservoirs Vulnerability Experiment (CARVE) campaign. They evaluated the polar WRF for May-Oct 2012 for v3.4.1 and v3.5.1 and Mar-Nov 2013 for v3.5.1 against surface observations of air temperature and wind speed and found good agreement. Barrow is on the of the focal points for validation in this study. They also found that the WRF-STILT transport framework identifies the top of the column enhancement to within 500m of the value identified by aircraft in 67% of the profiles used by Chang et al. (2014). WRF-STILT can capture the shape and approximate depth of the $CH_4$ enhancement throughout the column in that study. Miller et al.

(2016) states that the systematic uncertainty of the calculated surface influence from WRF-STILT is estimated at 10–20%.

However, Zona et al. (2016) note that WRF estimates of PBL ventilation rates are difficult to assess quantitatively and might be subject to particular bias in the fall (and winter), when heat fluxes are low. Additionally, the surface evaluation by Henderson et al. (2015) found the largest biases along the coast of the North Slope. We agree and acknowledge that small biases exist at different times and locations, especially during the transition seasons along the coast, but their influence is hard to quantify. Evaluation of transport errors in this region, especially in the presence of snow cover, requires future study. The discrepancies we see between observed and simulated $\Delta CO_2$ at BRW are supported by multiple years of data, and any errors in the transport does not discount the large differences between the cold season $CO_2$ flux models evaluated here, as errors are applied equally to all models.

The WRF-STILT description in Sect. 2.3 now reads as follows:
   "The footprints are generated by the Lagrangian atmospheric transport modeling system, WRF-STILT (Stochastic Time-Inverted Lagrangian Transport model driven by Weather Research and Forecasting model meteorology (Henderson et al., 2015)). In this system, WRF meteorological fields are first generated for the study region and time period (v3.5.1 for ARM-ACME V and NOAA BRW tower footprints used here, v3.9.1 for ABoVE Arctic-CAP footprints). STILT then uses the WRF meteorology to estimate the contribution of surface fluxes to the atmospheric concentration at a specified time and place, called a receptor, by calculating the amount of time air (represented by a distribution of particles) spends in the lower half of the boundary layer at a given location. The WRF-STILT model configurations from Henderson et al. (2015) have been used extensively in numerous previous papers to study greenhouse gas fluxes using observations from aircraft and towers in Alaska, including on the North Slope (e.g., Chang et al., 2014; Miller et al., 2016; Zona et al., 2016; Commane et al., 2017; Karion et al., 2015; Hartery et al., 2018). An evaluation by Henderson et al. (2015) for WRF v.3.4.1 and v3.5.1 showed that their polar WRF configuration performs well against surface observations of air temperature and wind speed in Alaska and that WRF-STILT can capture the shape and approximate depth of greenhouse gases in the column. Zona et al. (2016) note that WRF planetary boundary layer ventilation rates may be biased in the fall (and winter) when heat fluxes are low, but this error is difficult to assess quantitatively. For this study, we use receptors set to correspond with the tower and aircraft $CO_2$ concentration observations. The footprints (and their corresponding measurements) for these receptors sample air from throughout the North Slope but are concentrated more heavily toward the area around the NOAA BRW tower (Fig. 1c)."

Potential errors in the transport model are now mentioned in Sect. 4.3.2 as a source of uncertainty in need of further study as follows:
   "Using these atmospheric observations is uncertain, however, due to potential errors in the transport modeling, which are difficult to quantify. Atmospheric modeling of remote areas such as the Alaska North Slope requires further evaluation and improvement."

5. WRF model set up. There is no mention about details of the WRF model setup used to derive the atmospheric transport and surface sensitivity footprints applied in this study. What is the horizontal and vertical resolution of the WRF model used? What version is applied? How many spatial domains were used in the simulations? What physics options (e.g., schemes for long- and short-wave radiation, microphysics, convection, PBL, land surface, etc.) were selected for the model simulations? The differences in WRF setups can directly impact the accuracy of the model predictions.

The WRF configuration used in this study was initially extensively described and evaluated in Henderson et al. (2015) for v3.4.1 and v3.5.1. Those versions were used extensively in numerous previous papers to study greenhouse gas fluxes from aircraft and towers in Alaska, including on the North Slope (e.g., Chang et al. (2014), Miller et al. (2016), Karion et al. (2016), Hartery et al. (2016), Zona et al. (2016), Commane et al. (2017)). This paper uses both WRF v3.5.1 and a recent update to v3.9.1 as mentioned on lines 174-175 of the main text.

See response to 4 above for changes made to WRF-STILT description.

6. Line 399-410. Beyond the fact that it improves the comparison of simulated $\Delta CO_2$ values to observations at BRW, why is the constant 0.25 µmol m-2 s-1 zero-curtain emission source applied for October, which decreases to zero in December, chosen to add to TVPRM constrained estimates? Are there any past studies which could justify adding this value? Some justification needs to be provided for why these zero-curtain emission values were chosen.

As stated in the main text on line 402, the chosen zero-curtain flux value is within the observed variability of the IVO and CMDL sites during the early cold season and its reduction into December is consistent with these observations. This is a simplistic approximation that is meant to demonstrate the importance of adding this missing flux variability.

Clarification related to this point is incorporated below.

Also, more detail is needed to why the coastal tundra ecosystem parameterization was applied for inland aquatic fluxes. What inland water map was used to derive the location of all inland water bodies? Is lake ice phenology considered when estimating inland aquatic fluxes? How much $CO_2$ is estimated to be emitted, or absorbed, by lakes throughout the year using these methods? In reality, lakes will have very little open water interaction with the atmosphere in the cold season as they can be frozen in this region.

The inland water map is determined from the ABoVE LC map, as initially described in Supplement. Again, the use of coastal tundra fluxes for the inland aquatic areas is a simplistic approximation that is meant to demonstrate the importance of adding this missing flux. We do not mean to imply that all inland water responds as coastal tundra, but rather that portions of inland water areas (the edges) are more similar to tundra (non-zero flux) than using a zero-flux assumption for all water. Ice phenology is not considered, but it may be similar to that of the freeze-thaw behavior soils of the coastal tundra.

This section is revised to read as follows:

"To account for these processes, we first add an additional $CO_2$ flux with zero-curtain timing to our constrained $CO_2$ flux (TVPRM) member from both inland and coastal tundra areas that consists of 0.25 µmol m$^{-2}$ s$^{-1}$ for October with a reduction to zero by the end of December. This peak additional $CO_2$ flux is within the daily variability of the observed $CO_2$ flux at the IVO and CMDL eddy flux sites during the zero-curtain period (Fig. S9) and the reduction into December is consistent with these observations. The additional zero-curtain flux improves the ability of the model to reproduce the observed $\Delta CO_2$ at the NOAA BRW tower (slope = 0.46, $R^2$ = 0.41). We also apply the coastal tundra site ecosystem parameterization used in our constrained TVPRM member to all areas of inland water on the North Slope, which account for 4% of the domain according to the ABoVE LC map (Fig. S5) and were previously set to zero $CO_2$ flux. Representing these aquatic areas with biogenic $CO_2$ fluxes consistent with coastal tundra ecosystems is one simple way to bridge the terrestrial-aquatic gap in tundra ecosystem models, where portions of aquatic systems on the land-water gradient (i.e., the edges) may be more likely to respond to the environment as coastal tundra than with the zero-flux assumed by water area. The ice phenology for areas of inland water producing $CO_2$ flux is then considered to be similar to that of the freeze-thaw timing in coastal tundra soils. Adding these coastal tundra fluxes to inland water areas also improves the performance of our model (slope = 0.32, $R^2$ = 0.30 against NOAA BRW tower observations). The magnitude of additional zero-curtain flux suggested here and the portion of inland water represented with coastal tundra site parameterizations produce the best statistical comparison for a range of choices tested (Fig. S17)."

We also clarify in the discussion in Sect. 4.2:

"The simplistic approximations suggested here are not inconsistent with the existing uncertainties in tundra $CO_2$ flux modeling and demonstrate the importance of considering these additional $CO_2$ fluxes and their mechanisms for future study."

7. Line 444-446. Net Annual $CO_2$ flux. The largest annual uptake of $CO_2$ between 2012 and 2017 was during 2013 and 2015. What was different about these years compared to the others in this time period? Is there a strong correlation with soil/air temperature, precipitation, snowpack, etc.? How about wildfires? From first glance it appears that these two years had the most acreage burned by fires in Alaska during the time period studied here (https://uaf-iarc.org/alaskas-changingwildfire-environment/). The text describes that the balance of Rsoil, Rplant, and GPP control the overall biospheric $CO_2$ flux; however, some description of the controlling variables on interannual variability of net $CO_2$ flux in this region would improve the scientific impact of this study.

On the annual timescale, 2013 and 2015 were stronger net sinks and 2014 was the strongest net source. The growing season (May-Aug) timeframe in each year determines the net sign as the cold season (Sep-Apr) net $CO_2$ flux is relatively constant from year to year.

The summer of 2015 was very warm, dry, and sunny in Alaska and resulted in extreme biomass burning activity outside of the North Slope (see response to 2 above). According to our best TVPRM simulation with additional $CO_2$ fluxes, the North Slope growing season net uptake in 2015 is very strong. This strength is due to high $T_a$, PAR, and SIF resulting in very high GPP.

Fairly high $T_a$ and $T_s$ also result in high $R_{soil}$ and $R_{plant}$, respectively, but this is not enough to offset the very high GEE.

In contrast, the summer of 2014 was cool, wet, and cloudy, and the simulated North Slope growing season net uptake is very weak. $T_a$, PAR, and SIF, all drivers of GPP, are very low. Lower-than-normal $T_a$ also results in very low $R_{plant}$, but as with 2015, this is not enough to offset the extremely low uptake.

Growing season 2013 maintains moderately high GPP, but with moderately low $R_{plant}$ and very low $R_{soil}$. Extremely low Ts contributed to the very low $R_{soil}$, likely as a result of above-average lingering snowpack into May. This is a bit surprising given that the mean snowpack for the proceeding cold season (Sep-Apr) was not particularly deep. This result supports the importance of snowmelt timing for net carbon exchange.

Discussion of the drivers of interannual variability in growing season fluxes has been added to Sect 4.1 in response to this comment and the comments of Reviewer 2 as follows:

"The growing season of each year determines the sign of the regional annual net $CO_2$ flux during our study period, with 2013 and 2015 being strong net sinks and 2014 being the strongest net source. The relative magnitude of each component of the net $CO_2$ flux during the growing season (i.e., $R_{soil}$, $R_{plant}$, GPP) varies from year-to-year (Table S7) and helps explain the interannual variability in the net source or sink status of the North Slope. Growing season 2015 was very warm, dry, and sunny in Alaska and resulted in extreme biomass burning activity outside of the North Slope (Table S1). High regional mean $T_a$ and PAR (Table S8) and low accumulated precipitation (Table S9) in NARR confirm this was the case for North Slope as well, with high $T_a$ and PAR contributing to a very high GPP. The growing season SIF signal from the CSIF product, which determines the seasonal cycle and relative magnitude of photosynthetic activity, is also large in 2015 (Table S8), further enhancing GPP. This year and others with a larger GPP component of NEE correspond to growing seasons with stronger SIF signals, which is an indicator of increased productivity and consistent with previous studies (e.g., Magney et al., 2019; Sun et al., 2017). While fairly high $T_a$ and $T_s$ in 2015 also result in high $R_{soil}$ and $R_{plant}$, respectively, this elevated respiration is not enough to offset the very high GPP and results in a large net $CO_2$ sink. In contrast, the summer of 2014 was cool, wet, and cloudy, and the North Slope experienced very low $T_a$, PAR, and SIF signal, producing very low GPP. Lower-than-normal $T_a$ also results in very low $R_{plant}$, but as with 2015, this is not enough to offset the extremely low uptake resulting in a large net $CO_2$ source for 2014. In 2013, the other growing season with a strong net $CO_2$ sink, moderately high GPP combines with moderately low $R_{plant}$ and very low $R_{soil}$. Extremely low $T_s$ causes this very low $R_{soil}$, which, relative to moderate $T_a$ and PAR, is likely a result of above-average lingering snowpack into May (Table S9). This lingering snowpack is perhaps surprising given that the mean snowpack for the proceeding cold season was not particularly deep. The important impact that snow cover and the timing of snowmelt has on $T_s$ and carbon response in tundra ecosystems has been recently emphasized (e.g., Kim et al., 2021), and is supported by our work, which shows that the prevalence of snow in the spring may determine the sign of the regional net $CO_2$ for an entire year."

The following were added to the Supplement as Tables S7-S9:

Table S7. Alaska North Slope growing season (May–Aug) net $CO_2$ flux by component for the TVPRM Constrained + ZC and IW scenario for 2012–2017.

| Flux Component | 2012 | 2013 | 2014 | 2015 | 2016 | 2017 |
|---|---|---|---|---|---|---|
| $R_{soil}$ [TgC] | 18 | 16 | 17 | 18 | 18 | 17 |
| $R_{plant}$ [TgC] | 33 | 30 | 28 | 33 | 33 | 30 |
| GPP [TgC] | 69 | 71 | 60 | 77 | 71 | 68 |
| NEE [TgC] | -18 | -25 | -15 | -25 | -19 | -21 |

Table S8. Alaska North Slope growing season (May-Aug) mean TVPRM drivers used in the TVPRM Constrained + ZC and IW scenario for 2012–2017, where the mean uses model gridboxes where the total ABoVE LC ocean and other land fraction is less than 0.5 (see Fig. S5).

| Driver | 2012 | 2013 | 2014 | 2015 | 2016 | 2017 |
|---|---|---|---|---|---|---|
| NARR $T_a$ [°C] | 7.4 | 6.6 | 6.2 | 7.5 | 7.8 | 6.8 |
| NARR $T_{scale}$ | 0.67 | 0.61 | 0.58 | 0.65 | 0.65 | 0.58 |
| NARR $T_s$ [°C] | 2.6 | 0.68 | 1.3 | 2.4 | 2.7 | 1.5 |
| NARR PAR [$\mu$mol photon m$^{-2}$ s$^{-1}$] | 484 | 478 | 466 | 495 | 497 | 507 |
| CSIF SIF product [mW m$^{-2}$ nm$^{-1}$ sr$^{-1}$] | 0.17 | 0.18 | 0.16 | 0.19 | 0.18 | 0.18 |

Table S9. Alaska North Slope growing season (May-Aug) mean additional select NARR Variables for 2012–2017, where the mean uses model gridboxes where the total ABoVE LC ocean and other land fraction is less than 0.5 (see Fig. S5).

| Variable | 2012 | 2013 | 2014 | 2015 | 2016 | 2017 |
|---|---|---|---|---|---|---|
| NARR 3hr accum. precipitation [kg m$^{-2}$] | 0.19 | 0.21 | 0.20 | 0.15 | 0.16 | 0.16 |
| NARR soil moisture content [kg m$^{-2}$] | 688 | 745 | 755 | 747 | 733 | 734 |
| NARR snow depth [m] | 0.046 | 0.076 | 0.032 | 0.030 | 0.026 | 0.040 |
| NARR snow cover fraction [0-1] | 0.15 | 0.20 | 0.16 | 0.12 | 0.11 | 0.17 |
| NARR snow depth [m] during proceeding Sep-Apr | 0.42 | 0.35 | 0.36 | 0.38 | 0.35 | 0.38 |
| NARR snow cover fraction [0-1] during proceeding Sep-Apr | 0.81 | 0.78 | 0.79 | 0.83 | 0.87 | 0.78 |

8. What are the scientific advancements of this study? The work does a nice job of combining in situ and remote-sensing data and models to estimate the annual net $CO_2$ flux from the North Slope of Alaska. However, beyond the detailed description of how the TVPRM estimates were optimized to match atmospheric observations, what is the importance of the TVPRM model development? A near neutral net annual $CO_2$ flux for the North Slope is derived with TVPRM which is said to be consistent with past model ensemble estimates (Fisher et al., 2014), so this result is really only novel compared to some past estimates from Luus et al. (2017), Natali et al. (2019), and Watts et al. (2021) discussed in the text. An interesting finding is the TVPRM prediction of interannual variability of $CO_2$ fluxes in the region. The fact that the model suggested the net annual $CO_2$ flux changes between small sources and sinks is interesting. The study states that variability in uptake season strength drives this variability; however, what are the physiochemical variables driving these differences? Is it precipitation, snowpack, air/soil temperature, fires, etc.? There is a lot that could be studied here to improve the novel aspects of the work. Looking into these physiochemical drivers, and their control on net $CO_2$ fluxes, would really help the reader understand what controlling variables could drive future changes in this region. This was stated in the text to be an importance of this work but really isn't addressed here at all.

TVPRM is a tool to be used to explore potential relationships between observed site-level net $CO_2$ fluxes, environmental drivers, and scaling methods. The key results of the paper come from the atmospheric observations which evaluate both the TVPRM ensemble and the previous flux estimates mentioned above. When constrained to the atmospheric observations, TVPRM estimates net $CO_2$ fluxes much lower in the late cold season compared to previous estimates by Luus et al. (2017) and Natali and Watts et al (2019). Fluxes from Luus et al. (2017) and Natali and Watts et al (2019) are also shown to be much too high. The differences between these are enough to change the North Slope from a consistent net source of $CO_2$ to a variable net source and sink between years. These results are highlighted in the paper.

We have expanded the discussion of controlling variables for the interannual variability to improve importance of this work as a response to 7 above and the comments of Reviewer 2.

We have also added implications for the work and how it could be used to improve modeling studies to end of discussion in Sect. 3.4.2:
    "The large initial range of potential regional net $CO_2$ flux values we found for the Alaska North Slope indicates a large sensitivity to choices and assumptions made when scaling eddy flux observations from the site- to regional- scale. The most important of these choices are the representation of the upland tundra, particularly for the response of $R_{soil}$ to $T_s$ during the cold season, and the distribution of vegetation types throughout the domain. Future tundra $CO_2$ modeling efforts should focus on using site-level data that is the most consistent with regional-scale fluxes, rather than incorporating data from all available sites. Consistency and accuracy in classification schemes used in vegetation maps must also be addressed. As we have shown with the atmospheric observations, not all model scenarios have equal likelihood to be true, and the mean of the model ensemble is not necessarily the most likely or most consistent with the atmosphere. Using these atmospheric observations is uncertain, however, due to potential errors in the transport modeling, which are difficult to quantify. Atmospheric modeling of remote areas such as the Alaska North Slope requires further evaluation and improvement. Further, increasing

model temporal resolution should be considered as the importance of the zero-curtain and snow cover to the net $CO_2$ flux of tundra ecosystems is recognized, both of which vary on the order of days and weeks, rather than months."

9. Could the TVPRM model be used with future gridded predictions of meteorology, vegetation, hydrology, and other sources of information to predict future changes in the net $CO_2$ flux of the North Slope? If so, it is likely beyond this study to do so, but this should be discussed in the conclusions section of the text to increase the scientific impact of this work.

TVPRM is a relatively simple, yet accurate model for net biogenic $CO_2$ flux when tuned to present-day eddy flux data. The model could be used for future projections, however, it is not prognostic, and the flux-driver relationships in the rapidly warming Arctic ecosystems are changing so quickly that we would not assume accuracy into the future.

The following text is added near the end of Sect. 4.3:
        "TVPRM could be used with projections of meteorology and SIF to calculate the future net $CO_2$ balance for this region, but we caution against overuse of the model using current parameters, as the flux-driver relationships in the rapidly warming Arctic ecosystems are changing so quickly that we would not assume accuracy into the future."

10. Vegetation maps. A major finding in this work is that vegetation distributions and ecosystem type information is a controlling factor on the ability to accurately model $CO_2$ fluxes in this region. Are the three vegetation maps used in this study (CAVM, RasterCAVM, ABoVE LC) the only ones available for this region? If there are other vegetation maps available, why aren't they used in this study since it is very important for TVPRM $CO_2$ flux calculation accuracy? If there are no other maps of vegetation distributions and ecosystem type, how should CAVM, RasterCAVM, and ABoVE LC be improved to assist improvement in CO2 flux calculation accuracy?

We are not aware of additional vegetation maps available that are both spatially explicit and represent the distribution of tundra ecosystems. The use of three maps to demonstrate the importance of vegetation map accuracy is already a key result and an element not considered by most other studies, which only use one. In Sect 4.1, we highlight the need for "…improved vegetation mapping and classification schemes…". How this can be accomplished is the subject of additional work.

11. Could the results from TVPRM be compared to net $CO_2$ flux estimates from other terrestrial biosphere models (e.g., CASA, SiB4, Jules, Orchidee, etc.) in this region? Does TVPRM improve upon these established terrestrial biosphere models?

We looked into these comparisons, but we decided it is not appropriate or fair to compare TVPRM with the other biosphere models mentioned in their standard versions (i.e., those used in CMIP5) are at much coarser spatial resolutions and do not explicitly account for permafrost tundra biogenic activity. The CMIP5 model net $CO_2$ fluxes for this region are highly variable, and few were able to accurately capture the magnitude or seasonal cycle of the $CO_2$ flux on the North Slope. For larger scale studies, we would encourage the evaluation of those models.

**Minor Comments**

1. Line 28. Not sure what "top-down" observations at atmospheric $CO_2$ are. Do the authors mean top-down emission estimates using atmospheric observations? In Sect. 2.3 I think top-down observations of $CO_2$ enhancements ($\Delta CO_2$) is the correct way to use this term. This occurs at other locations throughout the manuscript. Observations of concentrations are themselves not typically classified as top-down, but enhancements and emission estimates using models and the observations are more often termed as top-down.

We thank the reviewer for suggesting this.

We have clarified the wording in the abstract and at the end of the introduction that top-down observations are of atmospheric concentration enhancements, rather than concentrations alone.

2. Line 78-9. IVO, CMDL, TVPRM, CSIF, and SIF have yet to be defined in the text.

These terms are introduced in the main text prior to the reference to Fig. 1d in Sect. 2.4. As the reviewer points out above, many of the captions are already quite long and additionally complicating this one with definitions does not seem appropriate.

3. Line 174-175. More appropriate to reference Lin et al. (2003) for WRF-STILT.

Henderson et al. (2015) is the appropriate reference for the WRF-STILT simulations used in this paper, as additionally clarified above. WRF and STILT are individually cited within that reference.

4. Figure S1. Are the multi-colored lines in each panel of Fig. S1 the "Lines for matching site parameters and locations are highlighted"? This needs to be described more clearly either in the figure caption or in the text. I had a very difficult time understanding what these lines represented.

The multi-colored lines in each panel are the results of the cross-site evaluation used to determine the site groupings for scaling from the site to regional level. In other words, in this figure, we run the model using each of the eight sets of site parameters using the meteorology and SIF inputs for each site location, for a total 64 parameter-input combinations.

We clarify the caption of Fig. S1 to read as follows:
"Timeseries of daily mean site-level net $CO_2$ flux for 2014 at eddy flux measurement sites on the Alaska North Slope (top left panel) used to determine TVPRM parameters. For the cross-site evaluation, each site panel uses the meteorology and SIF at that site to calculate the TVPRM simulated net $CO_2$ flux using the parameters determined for all sites, with the colored lines corresponding to the sites in the top left panel. Here we show TVPRM net $CO_2$ flux driven by NARR meteorology and the CSIF SIF product, where the net $CO_2$ flux for corresponding site parameters and locations are highlighted using lines with heavier weight. Black dots show observed net $CO_2$ flux at each site." We also add references to the cross-site evaluation for clarification in Sect. 2.4 of the main text.

5. Figure S4. This figure has a lot of information in it yet is only introduced in the text. Can the authors describe the performance of the model in more detail? A couple sentences discussing intersite performance and the differences between seasons and averaging time periods would be helpful as there are very large differences which would be of interest to the reader.

This figure and the site-level evaluation of TVPRM are already described in Sect. S4 of the Supplement, which is referenced in Sect. 2.4 of the main text. The placement of the discussion is appropriate given that the site-level performance of the model is not the highlight of the paper.

We now include a brief description of the intersite performance in Sect. S4: "Intersite performance is more variable compared to the model performance trends across seasons and timescales. The relative quality of model performance at each site is likely due to the data availability for that site for a given averaging length or timeframe."

6. Line 318-321. In Fig. 3b the reader can not distinguish between the early and late cold season as discussed in the text. Only in Fig. S11 is the temporal color scaled used.

Agreed.

Temporal color scale added in Figs. 3b–3c. Reference to Fig. 3c also added, since that panel is now similar to those referenced for comparison in Fig. S11. Figure caption edited as needed.

Revised Fig. 3:

[Figure]

7. Fig. S14. To compare TVPRM Constrained with RS-PM Tsoil to the TVPRM Constrained using NARR data the text needs to reference which figure and sub-panel to compare Fig. S14a to.

The sentence comparing TVPRM Constrained with RS-PM Tsoil to the TVPRM Constrained using NARR now refers to Figs. 4a, S12, S14a as needed: "A single layer of $T_s$ at 8 cm depth from RS-PM (Fig. S14a) captures the magnitude and temporal behavior of the observed early cold season CO2 fluxes slightly better than the constrained member (Figs. 4a, S12), which uses NARR reanalysis $T_s$ and does not incorporate permafrost-model derived $T_s$."

8. Line 398. Missing "the" in this sentence.

Agreed this sentence could be confusing.

Clarified to read:
   "None of the flux products discussed above, including our TVPRM ensemble, account for any potential $CO_2$ fluxes during the zero-curtain period that are not driven by $T_s$ or are from areas on the terrestrial-aquatic interface."

9. Line 56 and throughout. Why is Natali et al. (2019) referenced as Natali & Watts et al., 2019? Also, "&" and "and" are interchangeable used throughout the paper. Might be better to choose one for consistency.

Natali and Watts shared first-authorship of their paper and prefer it cited this way (personal communication).

All "&" have been changed to "and" for consistency.

10. Line 516. "motivate" instead of "motive".

Fixed.

[revised manuscript text omitted]

---

## Author Comment (AC2)

**RC2**

I reviewed this manuscript for a previous submission. This remains an extremely impressive and comprehensive model- and observation- based analysis of tundra carbon cycling. The authors go through a fairly exhaustive list of modeling scenarios in a valiant attempt to explain a very limited set of observed growing season and cold season emissions. The results provide a very nice analysis of different model representations of seasonal dco2 timing and magnitude.

We thank the reviewer for their helpful comments and suggestions. Specific responses to each comment follow in red, with proposed edits to the manuscript in blue. Line numbers refer to the original manuscript.

The discussion section hasn't changed much. It could still use more qualitative discussion of results, with more references to the literature (here are several paragraphs with no references) to help explain/support findings.

We have rearranged and expanded the discussion section as noted below and through response to Reviewer 1.

Given the large ensemble of model scenarios, I was hoping to see a more focused discussion of how these difference scenarios (ecosystem parameterization, vegetation distribution, meteorological inputs) affect regional carbon balance, as a way to characterize uncertain and inform future modeling efforts. These scenarios are discussed sporadically throughout, but I think it would help to add separate section to the Discussion summarizing these effects.

We agree that this discussion was lacking in the previous version.

We have added a summary of the scenarios and how they could be used to inform modeling studies to end of discussion in Sect. 3.4.2:

"The large initial range of potential regional net $CO_2$ flux values we found for the Alaska North Slope indicates a large sensitivity to choices and assumptions made when scaling eddy flux observations from the site- to regional- scale. The most important of these choices are the representation of the upland tundra, particularly for the response of $R_{soil}$ to $T_s$ during the cold season, and the distribution of vegetation types throughout the domain. Future tundra $CO_2$ modeling efforts should focus on using site-level data that is the most consistent with regional-scale fluxes, rather than incorporating data from all available sites. Consistency and accuracy in classification schemes used in vegetation maps must also be addressed. As we have shown with the atmospheric observations, not all model scenarios have equal likelihood to be true, and the mean of the model ensemble is not necessarily the most likely or most consistent with the atmosphere. Using these atmospheric observations is uncertain, however, due to potential errors in the transport modeling, which are difficult to quantify. Atmospheric modeling of remote areas such as the Alaska North Slope requires further evaluation and improvement. Further, increasing model temporal resolution should be considered as the importance of the zero-curtain and snow cover to the net $CO_2$ flux of tundra ecosystems is recognized, both of which vary on the order of days and weeks, rather than months."

L363-368: It's not clear why a "PF-Model Derived Soil Temperature" is required to more accurately capture soil freezing processes. Is this process unique to PF affected regions, or are there other factors at play related more generally to soil thermodynamics, hydraulic properties, freeze-thaw dynamics, etc?

The soil temperatures from the Remote Sensing-Permafrost Model (RS-PM) are an Alaska-specific data product developed for permafrost zones to better understand the impact of climate warming on soil carbon loss (Yi et al. (2018, 2019)). RS-PM uses more tailored inputs, derived specifically for Arctic Alaska, to determine soil temperatures than those used by global- and regional-scale reanalysis products such as NARR and ERA5. These input datasets include higher spatial-resolution snow depth and variable soil dielectric constants derived from airborne radar. The configurations and parameterizations in RS-PM were also developed and tested using soil temperature and active layer thickness measurements from the North Slope. Further, RS-PM produces soil temperatures at higher vertical resolution in the near-surface than the reanalysis products, which is important to capture the subsurface heterogeneity in unfrozen soil which may be responsible for continued soil respiration during the zero-curtain throughout the freezing and thawing time periods.

Although we found limited improvement in the TVPRM cold season net $CO_2$ fluxes compared to the atmospheric observations when we implemented RS-PM soil temperatures, it was important to test this Alaska-specific permafrost soil temperature product. Using soil temperature itself, rather than any specific soil temperature product, seems to be the limiting factor in reproducing the observed cold season net $CO_2$ fluxes.

We have re-written portions of the text to better reflect the above description of the RS-PM soil temperatures:
    in Sect. 2.4: "RS-PM uses tailored input for Alaska permafrost zones, such as downscaled snow depth and aircraft-observed soil dielectric constants and was developed and tested using $T_s$ and active layer thickness measurements from the North Slope. RS-PM also produces $T_s$ at higher vertical resolution in the near-surface than the reanalysis products to capture subsurface heterogeneity in unfrozen soil, which is important to represent the zero-curtain throughout the freezing and thawing periods in Alaska."
    in Sect. 3.2: "To test the impact of reanalysis $T_s$ on the early cold season $CO_2$ fluxes, we implement $T_s$ that are more specifically developed to represent Alaska tundra permafrost soils during freeze-thaw processes than the reanalysis products driving our constrained TPVRM member."

L373-374: Would more SOC, or more labile soil C (e.g., Jeong et al 2018), help to elevate fall soil C emission rates?

Since $R_{soil}$ in TVPRM is derived from the site-level eddy flux measurements, the impact of all forms of soil carbon on the emission rates are implicitly included in the formulation. There is not a way to explicitly add additional SOC or more labile soil C in the current model framework. Should the relationship between soil carbon and emissions change in the future (more carbon available to be respired) in a way not related to $T_s$, then the parameters calculated would no longer be accurate.

We now refer to this potential scenario in Sect. 4.3, which was added in response to a comment by Reviewer 1:
"TVPRM could be used with projections of meteorology and SIF to calculate the future net $CO_2$ balance for this region, but we caution against overuse of the model using current parameters, as the flux-driver relationships in the rapidly warming Arctic ecosystems are changing so quickly that we would not assume accuracy into the future."

L449-451: Could you please elaborate on the "expected" response of tundra ecosystems to light and heat/temperature?

The previous wording was unclear.

We have revised this sentence to read as follows:
"The good performance of the TVPRM ensemble against the atmospheric observations during the growing season indicates that the tundra ecosystems of the Alaska North Slope respond to light and heat as quantified by PAR, $T_s$, and $T_a$, and that the net $CO_2$ flux is largely controlled by the simple $R_{soil}$, $R_{plant}$, and GPP relationships in the empirical model over this time."

L452-459: It is interesting that coastal ecosystems are more representative of North Slope, due to increased sensitivity to light. Is this a statement of a specific vegetation type, or more general statement that north slope vegetation is more sensitive to light, for example as an adaptation to long dark cold seasons. This discussion really could use some references to the literature to support some of these claims. Also reading ahead to 596-608 suggests that "net flux" could also be affected by respiration due to topography and soil inundation. Could the authors please speculate on the competing roles of vegetation/GPP vs topography/soil water/TER on GS net flux?

We do not say that coastal ecosystems are more representative of the North Slope as a whole, but rather that our analysis suggests that the ecosystem response of the southern North Slope (away from the coast) is consistent with coastal ecosystems (lines 456-457), because vegetation maps with more coastal tundra in the southern North Slope produce more uptake for the same drivers and better match with the atmospheric observations. While the southern North Slope areas are more consistent with coastal tundra, it is possible that these areas are misclassified in either our simplified two-tundra type scheme or in the vegetation maps themselves.

In TVPRM, coastal tundra does take up more $CO_2$ for a given unit of PAR, which could be evidence for an adaptation to lower light levels. Figure S1 supports this claim, where we show that coastal tundra growing season uptake is very high (panels for IVO, ICS, ICH, ICT) when driven by inland (more southern) tundra site meteorology ($T_a$, PAR) and SIF. The λ parameter values reported by Luus et al. (2017) also indicate greater uptake at "wetland" sites like Atqasuk and Barrow than at "graminoid tundra" sites like Ivotuk and Imnavait when all driver inputs are constant. Further, Mbufong et al. (2014) found that peak growing season net uptake for constant light is also greater at Barrow than at Ivotuk. However, when considering the ability of coastal tundra to take up $CO_2$ when moved toward the south, Patankar et al. (2013) saw that tundra plants exposed to additional intense light did not respond with additional uptake.

This section has been modified to read:

"The regional net $CO_2$ flux is highly sensitive, however, to the distribution of tundra vegetation types (upland v. coastal) throughout the North Slope during the growing season. Coastal tundra takes up more $CO_2$ for a given unit PAR compared to inland tundra, based on the relationships between observed site-level net $CO_2$ flux and PAR in this study (TVPRM parameters, Fig. S1), which could be evidence for an adaptation to lower light levels. This difference is consistent with Luus et al. (2017), who calculated greater uptake at "wetland" sites like Atqasuk and Barrow than at "graminoid tundra" sites like Ivotuk and Imnavait when all driver inputs are constant and with Mbufong et al. (2014), who also found that peak growing season net uptake for constant light is greater at Barrow than at Ivotuk. The stronger $CO_2$ uptake response of coastal tundra to light is important to consider due to the fact that the vegetation distributions assessed here with more coastal tundra to the south (CAVM (Walker et al., 2005), ABoVE LC (Wang et al., 2020)) better agree with the atmospheric observations. When considering the ability of coastal tundra to take up $CO_2$ when moved toward the south, Patankar et al. (2013) saw that tundra plants exposed to additional intense light did not respond with additional uptake. Therefore, while the ecosystem response of the southern North Slope is more consistent with coastal ecosystems, it seems possible that these areas are misclassified in either our simplified two-tundra type scheme or in the vegetation maps themselves. The large variability in net $CO_2$ flux calculated by using the different maps supports the importance of accurate ecosystem type locations in upscaling eddy flux measurements and highlights the need for improved vegetation mapping and classification schemes in the Arctic ecology research community."

The section on respiration referred to on lines 473-482 points out the importance of topography and soil inundation as contributing factors to the Rsoil-$T_s$ relationships derived at the individual eddy flux sites. These relationships vary greatly between the eight sites, and we have tested each of them against the atmospheric observations to see which is most consistent with the response of the North Slope. Varying topography and soil inundation throughout the region means that each of the site relationships is likely to be representative of many different locations, but the regional-scale response seems to be most consistent with IVO for inland tundra and CMDL for coastal tundra.

This section has been modified to read:
"The largest differences in the net $CO_2$ flux between TVPRM ensemble members result from the contrasting site conditions driving the ICS and ICT $R_{soil}$ parameterizations during the cold season. When taken separately by cold season segment, ICS members perform quite well against observations at the NOAA BRW tower for early cold season and ICT members perform well for the late cold season. The contrasting performance between site parameterizations is due to the topographic and hydrologic conditions, which are quite heterogeneous over a short distance and influence the plant communities and carbon storage, at each site. The ecosystems sampled by the ICS tower are seasonally inundated and retain a deep layer of organic soil that can be respired in greater amounts longer into the early cold season, while the well-drained hillslope at ICT does not allow for accumulation of organic matter in the same way (Euskirchen et al., 2017; Larson et al., 2021). While varying topography and soil inundation throughout the North Slope means that each of these site relationships is likely to be representative of many other locations in the region with similar conditions, the early-to-late cold season reduction in $CO_2$ fluxes at these sites is not

consistent with the observed regional atmospheric trend, however, and we remove the members parameterized by them from the ensemble. Individual eddy flux site parameterizations may reproduce regional $CO_2$ fluxes for a given season, but it is important to consider their response to drivers across multiple seasons when scaling from the site-level to regional domains."

We have also expanded the discussion of competing roles of respiration and GPP on interannual variability in Sect 4.1 in response to this comment and those by Reviewer 1.

The interannual variability discussion now reads as follows:

"The growing season of each year determines the sign of the regional annual net $CO_2$ flux during our study period, with 2013 and 2015 being strong net sinks and 2014 being the strongest net source. The relative magnitude of each component of the net $CO_2$ flux during the growing season (i.e., $R_{soil}$, $R_{plant}$, GPP) varies from year-to-year (Table S7) and helps explain the interannual variability in the net source or sink status of the North Slope. Growing season 2015 was very warm, dry, and sunny in Alaska and resulted in extreme biomass burning activity outside of the North Slope (Table S1). High regional mean $T_a$ and PAR (Table S8) and low accumulated precipitation (Table S9) in NARR confirm this was the case for North Slope as well, with high $T_a$ and PAR contributing to a very high GPP. The growing season SIF signal from the CSIF product, which determines the seasonal cycle and relative magnitude of photosynthetic activity, is also large in 2015 (Table S8), further enhancing GPP. This year and others with a larger GPP component of NEE correspond to growing seasons with stronger SIF signals, which is an indicator of increased productivity and consistent with previous studies (e.g., Magney et al., 2019; Sun et al., 2017). While fairly high $T_a$ and $T_s$ in 2015 also result in high $R_{soil}$ and $R_{plant}$, respectively, this elevated respiration is not enough to offset the very high GPP and results in a large net $CO_2$ sink. In contrast, the summer of 2014 was cool, wet, and cloudy, and the North Slope experienced very low $T_a$, PAR, and SIF signal, producing very low GPP. Lower-than-normal $T_a$ also results in very low $R_{plant}$, but as with 2015, this is not enough to offset the extremely low uptake by GPP resulting in a large net $CO_2$ source for 2014. In 2013, the other growing season with a strong net $CO_2$ sink, moderately high GPP combines with moderately low $R_{plant}$ and very low $R_{soil}$. Extremely low $T_s$ causes this very low $R_{soil}$, which, relative to moderate $T_a$ and PAR, is likely a result of above-average lingering snowpack into May (Table S9). This lingering snowpack is perhaps surprising given that the mean snowpack for the proceeding cold season was not particularly deep. The important impact that snow cover and the timing of snowmelt has on $T_s$ and carbon response in tundra ecosystems has been recently emphasized (e.g., Kim et al., 2021), and is supported by our work which shows that the prevalence of snow in the spring may determine the sign of the regional net $CO_2$ for an entire year."

The following were added to the Supplement as Tables S7-S9:

Table S7. Alaska North Slope growing season (May–Aug) net $CO_2$ flux by component for the TVPRM Constrained + ZC and IW scenario for 2012–2017.

| Flux Component | 2012 | 2013 | 2014 | 2015 | 2016 | 2017 |
|---|---|---|---|---|---|---|
| $R_{soil}$ [TgC] | 18 | 16 | 17 | 18 | 18 | 17 |
| $R_{plant}$ [TgC] | 33 | 30 | 28 | 33 | 33 | 30 |
| GPP [TgC] | 69 | 71 | 60 | 77 | 71 | 68 |
| NEE [TgC] | -18 | -25 | -15 | -25 | -19 | -21 |

Table S8. Alaska North Slope growing season (May-Aug) mean TVPRM drivers used in the TVPRM Constrained + ZC and IW scenario for 2012–2017, where the mean uses model gridboxes where the total ABoVE LC ocean and other land fraction is less than 0.5 (see Fig. S5).

| Driver | 2012 | 2013 | 2014 | 2015 | 2016 | 2017 |
|---|---|---|---|---|---|---|
| NARR $T_a$ [°C] | 7.4 | 6.6 | 6.2 | 7.5 | 7.8 | 6.8 |
| NARR $T_{scale}$ | 0.67 | 0.61 | 0.58 | 0.65 | 0.65 | 0.58 |
| NARR $T_s$ [°C] | 2.6 | 0.68 | 1.3 | 2.4 | 2.7 | 1.5 |
| NARR PAR [µmol photon $m^{-2}$ $s^{-1}$] | 484 | 478 | 466 | 495 | 497 | 507 |
| CSIF SIF product [mW $m^{-2}$ $nm^{-1}$ $sr^{-1}$] | 0.17 | 0.18 | 0.16 | 0.19 | 0.18 | 0.18 |

Table S9. Alaska North Slope growing season (May-Aug) mean additional select NARR Variables for 2012–2017, where the mean uses model gridboxes where the total ABoVE LC ocean and other land fraction is less than 0.5 (see Fig. S5).

| Variable | 2012 | 2013 | 2014 | 2015 | 2016 | 2017 |
|---|---|---|---|---|---|---|
| NARR 3hr accum. precipitation [kg $m^{-2}$] | 0.19 | 0.21 | 0.20 | 0.15 | 0.16 | 0.16 |
| NARR soil moisture content [kg $m^{-2}$] | 688 | 745 | 755 | 747 | 733 | 734 |
| NARR snow depth [m] | 0.046 | 0.076 | 0.032 | 0.030 | 0.026 | 0.040 |
| NARR snow cover fraction [0-1] | 0.15 | 0.20 | 0.16 | 0.12 | 0.11 | 0.17 |
| NARR snow depth [m] during proceeding Sep-Apr | 0.42 | 0.35 | 0.36 | 0.38 | 0.35 | 0.38 |
| NARR snow cover fraction [0-1] during proceeding Sep-Apr | 0.81 | 0.78 | 0.79 | 0.83 | 0.87 | 0.78 |

L460-465: This paragraph basically says that net uptake increases sometimes because of SIF, but we don't know why. I think more effort is needed to explain why. If its not because of air temperature or PAR, could it be soil temp? soil moisture? longer growing season? Different freeze/thaw dynamics?

We agree that more description was required to explain the net uptake increases.

We have expanded discussion of the variability of drivers leading to interannual variability in net uptake in Sect 4.1 in response to this comment and those by Reviewer 1. The new text of this discussion is copied above.

L493-495: Please elaborate on the processes driving the "physical release of CO2 from soil." I'm confused what could be the source of carbon if not from microbial activity. Please also comment on the possible role of emissions from permafrost and talik.

$CO_2$ produced by microbial activity in the soil must be released into the atmosphere before counted as an emissions source. When $CO_2$ is trapped between frozen/freezing layers or under the snowpack, there will be a disconnect between the microbial production rate of $CO_2$ and the emission rate of $CO_2$ into the atmosphere. The addition of the zero-curtain (ZC) emissions accounts for the observed sporadic delayed release of $CO_2$ produced when $T_s$ was higher.

We have modified this section to read:
"The additional zero-curtain flux represents large-scale emission events not directly timed to microbial activity and root respiration controlled by $T_s$, but could be related to the delayed physical release of previously produced $CO_2$ from soil through the snowpack as the soil layers remain unfrozen (Bowling and Massman, 2011)."

L516-528: It's surprising to see no mention of existing or future satellite datasets, which are getting better at resolving cold season emissions (e.g., Byrne et al., 2022)

Satellite products that rely on reflected sunlight such as $XCO_2$ from OCO-2 have essentially no coverage on the North Slope from October to March (Byrne et al., 2022). Inversions using only $XCO_2$ that cover this time period would be influenced by observations from farther south, where $CO_2$ emissions are more likely to continue into the cold season.

We now mention the limitations of satellite datasets during the cold season in Sect. 4.3.1:
"Satellites that rely on reflected sunlight to detect $CO_2$ have increasingly been used to constrain $CO_2$ budgets in the northern latitudes (e.g., Byrne et al., (2022)), but data is very limited in the cold season, especially in far-northern regions like the North Slope."

Jeong, S. J., Bloom, A. A., Schimel, D., Sweeney, C., Parazoo, N. C., Medvigy, D., … Miller, C. E. (2018). Accelerating rates of arctic carbon cycling revealed by long-term atmospheric CO2 measurements. Science Advances, 4(7), 1–7. https://doi.org/10.1126/sciadv.aao116

Byrne, B., Liu, J., Yi, Y., Chatterjee, A., Basu, S., Cheng, R., Doughty, R., Chevallier, F., Bowman, K. W., Parazoo, N. C., Crisp, D., Li, X., Xiao, J., Sitch, S., Guenet, B., Deng, F., Johnson, M. S., Philip, S., McGuire, P. C., and Miller, C. E.: Multi-year observations reveal a larger than expected autumn respiration signal across northeast Eurasia, Biogeosciences, 19, 4779–4799, https://doi.org/10.5194/bg-19-4779-2022, 2022.

**Response References**
Bowling, D. R. and Massman, W. J.: Persistent wind-induced enhancement of diffusive CO2 transport in a mountain forest snowpack, J. Geophys. Res. Biogeosci., 116, G04006, https://doi.org/10.1029/2011JG001722, 2011.

Euskirchen, E. S., Bret-Harte, M. S., Shaver, G. R., Edgar, C. W., and Romanovsky, V. E.: Long-Term Release of Carbon Dioxide from Arctic Tundra Ecosystems in Alaska, Ecosystems, 20, 960–974, https://doi.org/10.1007/s10021-016-0085-9, 2017.

Kim, J., Kim, Y., Zona, D., Oechel, W., Park, S.-J., Lee, B.-Y., Yi, Y., Erb, A., and Schaaf, C. L.: Carbon response of tundra ecosystems to advancing greenup and snowmelt in Alaska, Nat Commun, 12, 6879, https://doi.org/10.1038/s41467-021-26876-7, 2021.

Larson, E. J. L., Schiferl, L. D., Commane, R., Munger, J. W., Trugman, A. T., Ise, T., Euskirchen, E. S., Wofsy, S., and Moorcroft, P. M.: The changing carbon balance of tundra ecosystems: results from a vertically-resolved peatland biosphere model, Environ. Res. Lett., 17, 014019, https://doi.org/10.1088/1748-9326/ac4070, 2021.

Luus, K. A., Commane, R., Parazoo, N. C., Benmergui, J., Euskirchen, E. S., Frankenberg, C., Joiner, J., Lindaas, J., Miller, C. E., Oechel, W. C., Zona, D., Wofsy, S., and Lin, J. C.: Tundra photosynthesis captured by satellite-observed solar-induced chlorophyll fluorescence, Geophys. Res. Lett., 44, 2016GL070842, https://doi.org/10.1002/2016GL070842, 2017.

Magney, T. S., Bowling, D. R., Logan, B. A., Grossmann, K., Stutz, J., Blanken, P. D., Burns, S. P., Cheng, R., Garcia, M. A., Köhler, P., Lopez, S., Parazoo, N. C., Raczka, B., Schimel, D., and Frankenberg, C.: Mechanistic evidence for tracking the seasonality of photosynthesis with solar-induced fluorescence, Proceedings of the National Academy of Sciences, 116, 11640–11645, https://doi.org/10.1073/pnas.1900278116, 2019.

Mbufong, H. N., Lund, M., Aurela, M., Christensen, T. R., Eugster, W., Friborg, T., Hansen, B. U., Humphreys, E. R., Jackowicz-Korczynski, M., Kutzbach, L., Lafleur, P. M., Oechel, W. C., Parmentier, F. J. W., Rasse, D. P., Rocha, A. V., Sachs, T., van der Molen, M. K., and Tamstorf, M. P.: Assessing the spatial variability in peak season $CO_2$ exchange characteristics across the Arctic tundra using a light response curve parameterization, Biogeosciences, 11, 4897–4912, https://doi.org/10.5194/bg-11-4897-2014, 2014.

Patankar, R., Mortazavi, B., Oberbauer, S. F., and Starr, G.: Diurnal patterns of gas-exchange and metabolic pools in tundra plants during three phases of the arctic growing season, Ecology and Evolution, 3, 375–388, https://doi.org/10.1002/ece3.467, 2013.

Sun, Y., Frankenberg, C., Wood, J. D., Schimel, D. S., Jung, M., Guanter, L., Drewry, D. T., Verma, M., Porcar-Castell, A., Griffis, T. J., Gu, L., Magney, T. S., Köhler, P., Evans, B., and Yuen, K.: OCO-2 advances photosynthesis observation from space via solar-induced chlorophyll fluorescence, Science, 358, eaam5747, https://doi.org/10.1126/science.aam5747, 2017.

Yi, Y., Kimball, J. S., Chen, R. H., Moghaddam, M., Reichle, R. H., Mishra, U., Zona, D., and Oechel, W. C.: Characterizing permafrost active layer dynamics and sensitivity to landscape spatial heterogeneity in Alaska, Cryosphere, 12, 145–161, https://doi.org/10.5194/tc-12-145-2018, 2018.

Yi, Y., Kimball, J. S., Chen, R. H., Moghaddam, M., and Miller, C. E.: Sensitivity of active-layer freezing process to snow cover in Arctic Alaska, Cryosphere, 13, 197–218, https://doi.org/10.5194/tc-13-197-2019, 2019.

Walker, D. A., Raynolds, M. K., Daniëls, F. J. A., Einarsson, E., Elvebakk, A., Gould, W. A., Katenin, A. E., Kholod, S. S., Markon, C. J., Melnikov, E. S., Moskalenko, N. G., Talbot, S. S., Yurtsev, B. A. (†), and Team, T. other members of the C.: The Circumpolar Arctic vegetation map, J. Veg. Sci., 16, 267–282, https://doi.org/10.1111/j.1654-1103.2005.tb02365.x, 2005.

Wang, J. A., Sulla-Menashe, D., Woodcock, C. E., Sonnentag, O., Keeling, R. F., and Friedl, M. A.: Extensive land cover change across Arctic–Boreal Northwestern North America from disturbance and climate forcing, Global Change Biol., 26, 807–822, https://doi.org/10.1111/gcb.14804, 2020.